# Transgenerational inheritance of centromere identity requires the CENP-A N-terminal tail in the *C. elegans* maternal germ line

**Reinier F. Prosée**[1], **Joanna M. Wenda**[1], **Isa Özdemir**[1], **Caroline Gabus**[1], **Kamila Delaney**[1], **Francoise Schwager**[2], **Monica Gotta**[2], **Florian A. Steiner**[1] *

**1** Department of Molecular Biology and Institute of Genetics and Genomics in Geneva, Section of Biology, Faculty of Sciences, University of Geneva, Geneva, Switzerland, **2** Department of Cell Physiology and Metabolism and Institute of Genetics and Genomics in Geneva, Faculty of Medicine, University of Geneva, Geneva, Switzerland

\* florian.steiner@unige.ch

**Data Availability Statement:** All relevant data are within the paper and its Supporting Information files.

## Abstract

Centromere protein A (CENP-A) is a histone H3 variant that defines centromeric chromatin and is essential for centromere function. In most eukaryotes, CENP-A-containing chromatin is epigenetically maintained, and centromere identity is inherited from one cell cycle to the next. In the germ line of the holocentric nematode *Caenorhabditis elegans*, this inheritance cycle is disrupted. CENP-A is removed at the mitosis-to-meiosis transition and is reestablished on chromatin during diplotene of meiosis I. Here, we show that the N-terminal tail of CENP-A is required for the de novo establishment of centromeres, but then its presence becomes dispensable for centromere maintenance during development. Worms homozygous for a CENP-A tail deletion maintain functional centromeres during development but give rise to inviable offspring because they fail to reestablish centromeres in the maternal germ line. We identify the N-terminal tail of CENP-A as a critical domain for the interaction with the conserved kinetochore protein KNL-2 and argue that this interaction plays an important role in setting centromere identity in the germ line. We conclude that centromere establishment and maintenance are functionally distinct in *C. elegans*.

## Introduction

Centromeres are crucial for the segregation of chromosomes into the daughter cells during cell division. They hold the sister chromatids together after DNA replication and serve as the sites of kinetochore formation. During mitosis, the kinetochore attracts and binds microtubules that pull the sister chromatids toward opposing spindle poles. In most species, the centromere is defined by the histone variant centromere protein A (CENP-A), which forms the chromatin foundation for building the kinetochore and acts as an epigenetic mark to maintain centromere identity [1,2]. CENP-A is almost universally conserved but differs in amino acid sequence between different taxa, especially in the N-terminal tail [3–6]. The C-terminal histone fold domain (HFD) is more conserved, since it incorporates into the histone octamer and is therefore under structural constraints [7].

**Funding:** The work was financially supported by the Swiss National Science Foundation (www.snf. ch, Grants 31003A_156774 and 31003A_175606 to FAS), and funding from the Republic and Canton of Geneva (to FAS and MG). The funders had no role in study design, data collection and analysis, decision to publish, or preparation of the manuscript.

**Competing interests:** The authors have declared that no competing interests exist.

**Abbreviations:** 3AT, 3-amino-1,2,4-triazole; AID, auxin-inducible degron; CATD, centromere targeting domain; CENP-A, centromere protein A; ChEC, chromatin endogenous cleavage; CTF, corrected total fluorescence; emb, embryonic lethal; FL, full-length; HFD, histone fold domain; IF, immunofluorescence; IPTG, isopropyl-β-D-thiogalactopyranoside; knl, kinetochore-null; KO, knockout; mel, maternal effect lethal; mRNA, messenger RNA; NEBD, nuclear envelope breakdown; PRC2, polycomb repressive complex 2; PTM, posttranslational modification; RNAi, RNA interference; RT, room temperature; TEV, tobacco etch virus; Y2H, yeast two-hybrid.

The incorporation and maintenance of CENP-A at the centromere is a highly regulated process [8,9]. During DNA replication, CENP-A levels are diluted by half and have to be replenished with newly synthesized CENP-A [10]. Most studies investigating CENP-A loading have been performed in cultured mitotic cells, where it is possible to tightly regulate the cell cycle. Although the mechanistic details differ between different species, the basic components for CENP-A loading and maintenance in mitosis are well conserved [2]. CENP-C and the MIS18 complex recognize CENP-A nucleosomes and are required for priming centromeric sites for new CENP-A deposition [11–14]. CENP-A–specific histone chaperones (HJURP/ CAL1/Scm3) then incorporate new CENP-A at centromeres, thereby maintaining CENP-A on chromatin [15–17]. Placeholder nucleosomes, containing histone H3 or the histone variant H3.3, can mark the location where new CENP-A will be loaded [3,18], and active transcription displaces these nucleosomes to allow for the deposition of the CENP-A nucleosomes [19–21].

How centromeres are inherited from one generation to the next in multicellular organisms is less well understood [22]. However, the inheritance of CENP-A protein into the zygote is crucial for centromere functioning in the next generation prior to the onset of zygotic transcription. This inheritance is achieved either through stability of the CENP-A nucleosomes or active recycling of CENP-A during meiosis [23–25]. In organisms that inherit CENP-A protein only on the paternal (e.g., *Arabidopsis thaliana*) or the maternal (e.g., *Caenorhabditis elegans*) chromosomes, the "template-governed" model of centromere inheritance has been disputed, because the paternal or maternal chromosomes devoid of CENP-A need to establish centromeres de novo after fertilization and prior to the first embryonic cell division [26–28]. An additional interruption of the templated inheritance of CENP-A occurs in the adult hermaphrodite germ line of *C. elegans*, where CENP-A is removed from chromatin at the mitosis-to-meiosis transition and reloaded during diplotene of meiosis I [27]. How this de novo establishment of CENP-A is regulated remains unclear.

*C. elegans* chromosomes are holocentric, with CENP-A distributed along the length of the entire chromosomes [29]. CENP-A is essential for mitotic cell divisions in the distal germ line and during development, but dispensable for chromosome segregation during meiosis [27]. Chromatin association of CENP-A in mitotic cells depends on the MIS18BP1 homolog KNL-2 [30] and on the general histone chaperone LIN-53, a homolog of RbAp46/48 [31]. The functional *C. elegans* CENP-A homolog is called HCP-3, but in this study, we will refer to it as CENP-A for clarity.

In this study, we demonstrate that the N-terminus of CENP-A mediates interaction with KNL-2 in vitro and in vivo. Surprisingly, removing the N-terminal tail of CENP-A does not affect centromere function during development, and the kinetochore-null (*knl*) phenotype associated with loss of centromeres only appears at the first embryonic division of the next generation. We find that correctly established centromere identity, rather than the presence of the full-length CENP-A protein, is essential for mitosis during development. Our results suggest that in *C. elegans*, the CENP-A N-terminal tail has acquired an important function in reestablishing the centromere in each generation in the adult germ line, but then its presence becomes dispensable for centromere maintenance during development.

## Results

### The CENP-A N-terminal tail interacts with the central domain of KNL-2

In *C. elegans*, CENP-A is chromatin associated in dividing cells throughout development but is removed and reestablished, respectively, during different stages of the adult hermaphrodite germ line [27] (Fig 1). In embryos, KNL-2 and CENP-A colocalize and are interdependent for chromatin association, and depletion of either KNL-2 or CENP-A leads to a *knl* phenotype

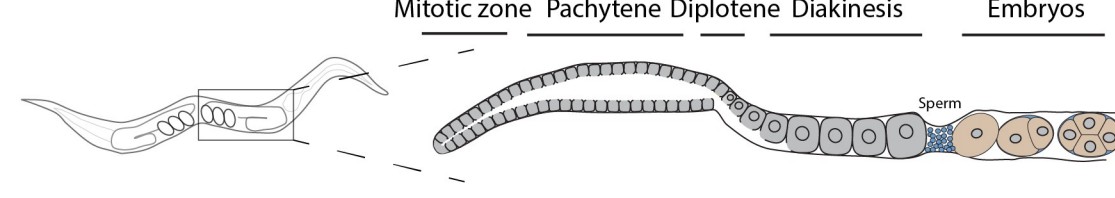

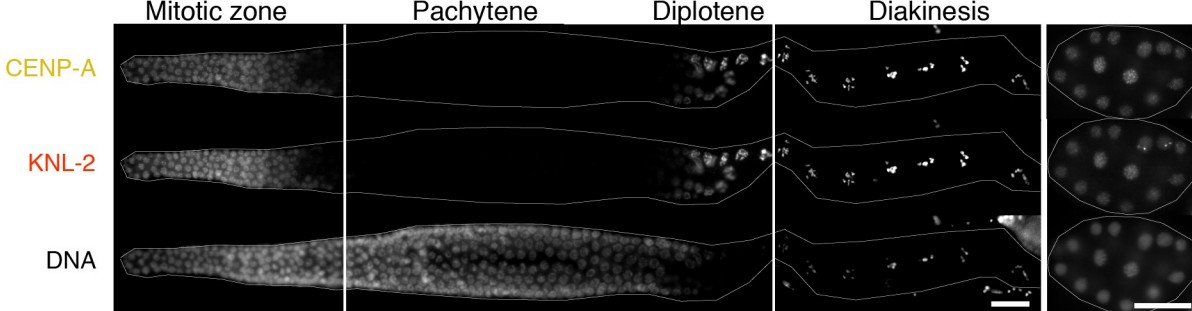

**Fig 1. Dynamics of KNL-2 and CENP-A in the *C. elegans* germ line.** Top, cartoon images of an adult hermaphrodite *C. elegans* and the different stages of the germ line. Bottom, IF images of CENP-A and KNL-2 in adult hermaphrodite germ lines and embryos, showing the removal at the mitosis-to-meiosis transition and the reappearance at the diplotene stage. The white lines indicate where images of the same gonad have been merged. Scale bars represent 20 μm. CENP-A, centromere protein A; IF, immunofluorescence.

[30,32]. CENP-A and KNL-2 also show overlapping germline distributions (Fig 1). We therefore hypothesized that KNL-2 may play a role in the de novo establishment of centromeres in the germ line.

We first investigated the interplay between CENP-A and KNL-2. We used a yeast two-hybrid (Y2H) assay to determine the minimal regions of both proteins that are required for their interaction. The interaction between full-length CENP-A and KNL-2 is barely detectable (Fig 2A, left), possibly due to improper folding of the proteins. Next, we tested the CENP-A HFD or the N-terminal tail in combination with full-length KNL-2. While we observed no interaction between the CENP-A HFD and KNL-2, the interaction between the CENP-A N-terminal tail and KNL-2 supported yeast growth on selective medium (SD-LTH containing 5 mM 3-amino-1,2,4-triazole (3AT); Fig 2A, left). We then tested constructs covering the N-terminal (aa 1–268), central (aa 269–470), and C-terminal (aa 471–877) regions of KNL-2. We only observed yeast growth on selective medium for the central region (aa 269–470), as was reported in an earlier study [33] (Fig 2A, middle). This part of KNL-2 does not contain any annotated domains and is predicted to be mostly unstructured. We next tested different parts of the CENP-A N-terminal tail for interaction with the KNL-2 central region and found that aa 1–122 are sufficient to promote yeast growth on selective medium (Fig 2A, right). We confirmed the interaction of the CENP-A N-terminal tail with the KNL-2 central region in vitro using recombinant proteins (S1A Fig).

Next, we tested the interaction of the CENP-A N-terminal tail with KNL-2 in vivo. We removed the part of the endogenous CENP-A gene encoding the first 99 amino acids using the CRISPR/Cas-9 system, creating CENP-A$^{aa100–287}$ (called CENP-A Δ-tail for simplicity). We maintained this truncation allele heterozygous over the wild-type allele in the context of a genetic balancer. In this balancer strain, the balanced allele with the full-length copy of

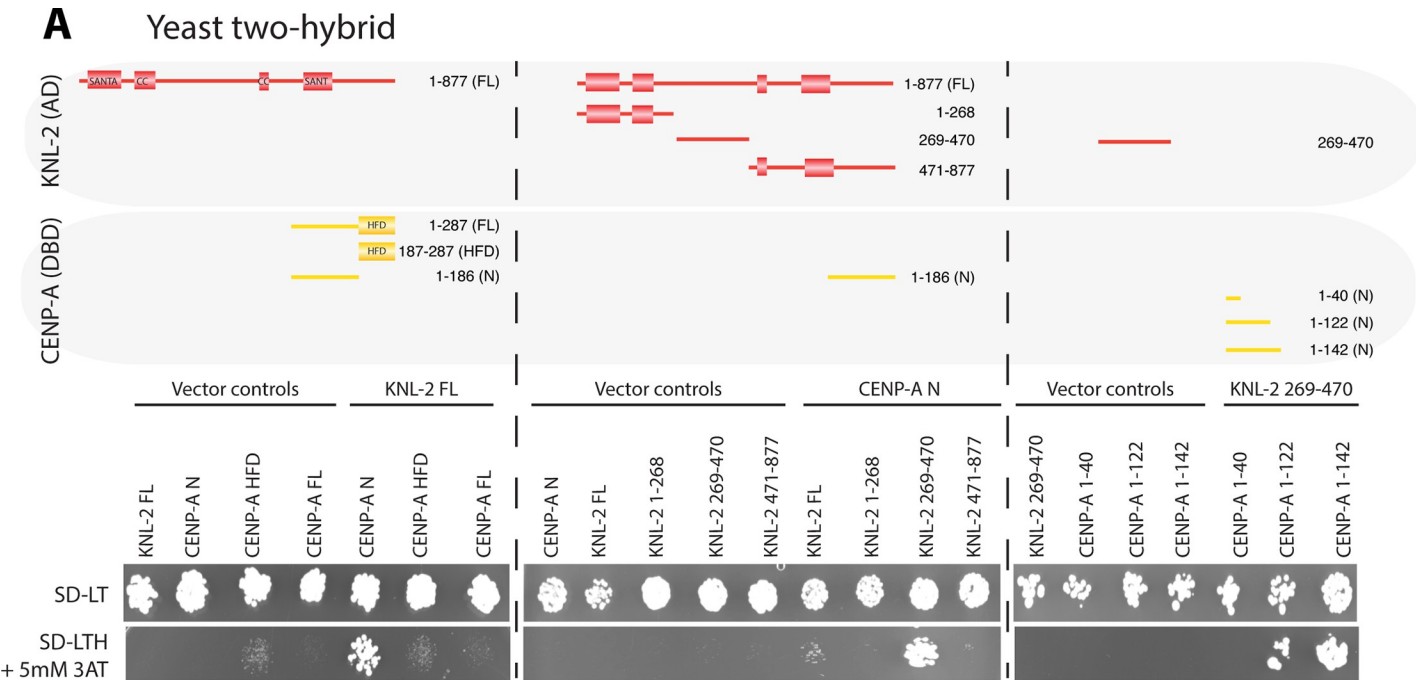

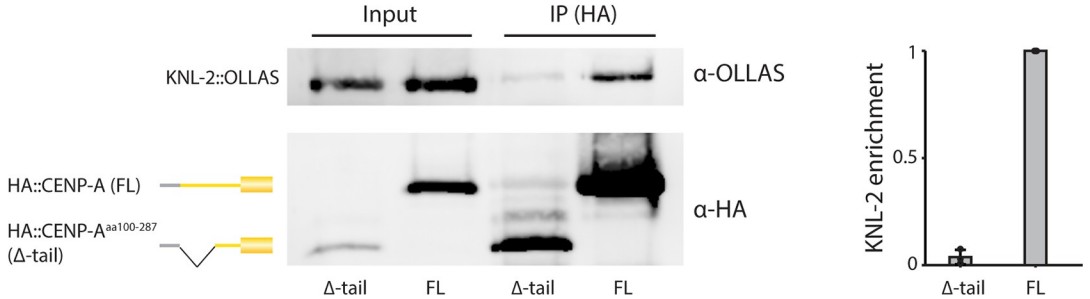

**Fig 2. The CENP-A N-terminal tail interacts with the KNL-2 central domain.** (A) Yeast two-hybrid analysis of the interaction between CENP-A and KNL-2. Cartoons of the different CENP-A and KNL-2 fragments used (top), showing the annotated SANTA, SANT/Myb, and CC domains in KNL-2 and HFD in CENP-A. The interaction between the constructs were determined by assaying growth of yeast cells on selective medium (SD-LTH, containing 5 mM 3AT). Vector-only constructs and nonselective plates (SD-LT) were used as negative and positive controls for growth, respectively. Left, full-length KNL-2 interaction with different parts of CENP-A. Center, interaction of the CENP-A N-terminal tail with different regions of KNL-2. Right, interaction of different CENP-A N-terminal tail fragments with the KNL-2 central region. (B) Co-IP experiments with KNL-2 and CENP-A FL or CENP-A$^{aa100-287}$ (Δ-tail). In cartoons, CENP-A is shown in yellow and HA in gray. Representative western blot showing IPs of HA-tagged CENP-A FL, or Δ-tail from embryonic extracts. HA-tagged constructs were IPed and detected using an anti-HA antibody. OLLAS-tagged KNL-2 was detected using an anti-OLLAS antibody in input and IP samples. The bar graph shows a quantification of relative KNL-2 enrichment, which was determined as the IP ratio of KNL-2/CENP-A divided by the input ratio of KNL-2/CENP-A, with the interaction for CENP-A FL set to 1 in each experiment. $N$ = 3 independent IP/western blotting experiments. The data underlying all the graphs can be found in S1 Data. CC, coiled-coil; CENP-A, centromere protein A; Co-IP, co-immunoprecipitation; FL, full-length; HFD, histone fold domain.

CENP-A (referred to as FL) is linked to a transgene expressing pharyngeal GFP, thus allowing us to visually distinguish worms heterozygous and homozygous for CENP-A Δ-tail. *C. elegans* populations mostly consist of hermaphrodites that produce both oocytes and sperm and self-fertilize to give rise to the next generation, although males that only produce sperm and rely

on cross-fertilization of hermaphrodites to give rise to offspring also exist. In this study, we use the heterozygous CENP-A FL/Δ-tail hermaphrodites as parental generation (P0) and label the first and second subsequent offspring generations as F1 and F2, respectively. In the balancer strain, offspring homozygous for CENP-A FL are embryonic lethal due to aneuploidy of the balanced chromosomes. We used this balancer strain to investigate the interaction of KNL-2 with CENP-A FL and Δ-tail in vivo. IP of the CENP-A Δ-tail protein only weakly co-precipitated KNL-2, while KNL-2 was easily detectable in IPs of CENP-A FL protein (Figs 2B and S1B).

We conclude that the CENP-A N-terminal tail is both necessary and sufficient for the interaction with KNL-2 (Figs 2 and S1). Based on our Y2H assay and the in vitro experiments, the central region of KNL-2 mediates this interaction (Figs 2A and S1A). These observations have recently been independently confirmed [34].

## The presence of the CENP-A N-terminal tail is dispensable during development

CENP-A and KNL-2 are both essential proteins, and loss-of-function mutations or depletion by RNAi result in early embryonic lethality due to chromosome segregation errors [30,35]. Based on the previously reported interdependence of CENP-A and KNL-2 in maintaining the centromere [30], we expected to observe an embryonic lethal (*emb*) phenotype when interrupting their interaction. However, we found that in the F1 generation, hermaphrodites homozygous for CENP-A Δ-tail were fully viable (Fig 3A). These worms looked and developed indistinguishably from wild-type worms and showed no obvious germline proliferation defects. In addition, homozygous F1 CENP-A Δ-tail males were viable and fertile. However, the F2 embryos laid by the homozygous F1 hermaphrodites were all inviable due to severe chromosome segregation defects (Fig 3A and 3B). Instead of the expected F1 *emb* phenotype, the CENP-A N-terminal tail deletion thus causes a maternal effect lethal (*mel*) phenotype.

In *C. elegans*, the CENP-A protein is exclusively inherited through the maternal germ line [27,28]. The *mel* phenotype caused by the CENP-A N-terminal tail deletion could potentially be explained by maternal contribution of the CENP-A FL protein from the P0 adults to the F1 embryos, as the maternal germ line (P0) is heterozygous for the N-terminal tail deletion. This would require the maternal CENP-A FL load to support the cell divisions during embryonic and larval stages as well as the germline proliferation in late larval and adult stages of the CENP-A Δ-tail F1 homozygotes. However, this is unlikely to be the case, since a strain containing a CENP-A knockout (KO) allele (in the context of the same genetic balancer) produces homozygous F1 offspring with an *emb* phenotype (Fig 3A), even though these embryos receive the same load of CENP-A FL protein from the maternal heterozygous germ line as those produced by the CENP-A Δ-tail strain (as described below).

We directly measured the extent of maternal CENP-A FL protein contribution and the onset of zygotic CENP-A expression in different genetic backgrounds (S2 Fig). We first crossed males with GFP- and HA-tagged CENP-A FL to feminized *fem-2* hermaphrodites, ensuring that all offspring are heterozygous for tagged CENP-A FL. Since CENP-A protein is not inherited paternally [27,28], all embryonic tagged CENP-A FL protein must be zygotically expressed, and we found that it was first detectable during early gastrulation (30 to 40 cell stage) (S2A Fig, top). We then let the heterozygous tagged CENP-A FL hermaphrodites self-fertilize and identified the limits of the maternal CENP-A contribution. We found that maternally contributed tagged CENP-A FL protein became undetectable mid-gastrulation (100 to 200 cell stage) in embryos without zygotic expression of the tagged CENP-A FL protein (S2A Fig, bottom). We repeated the analysis of maternal contribution with worms heterozygous for

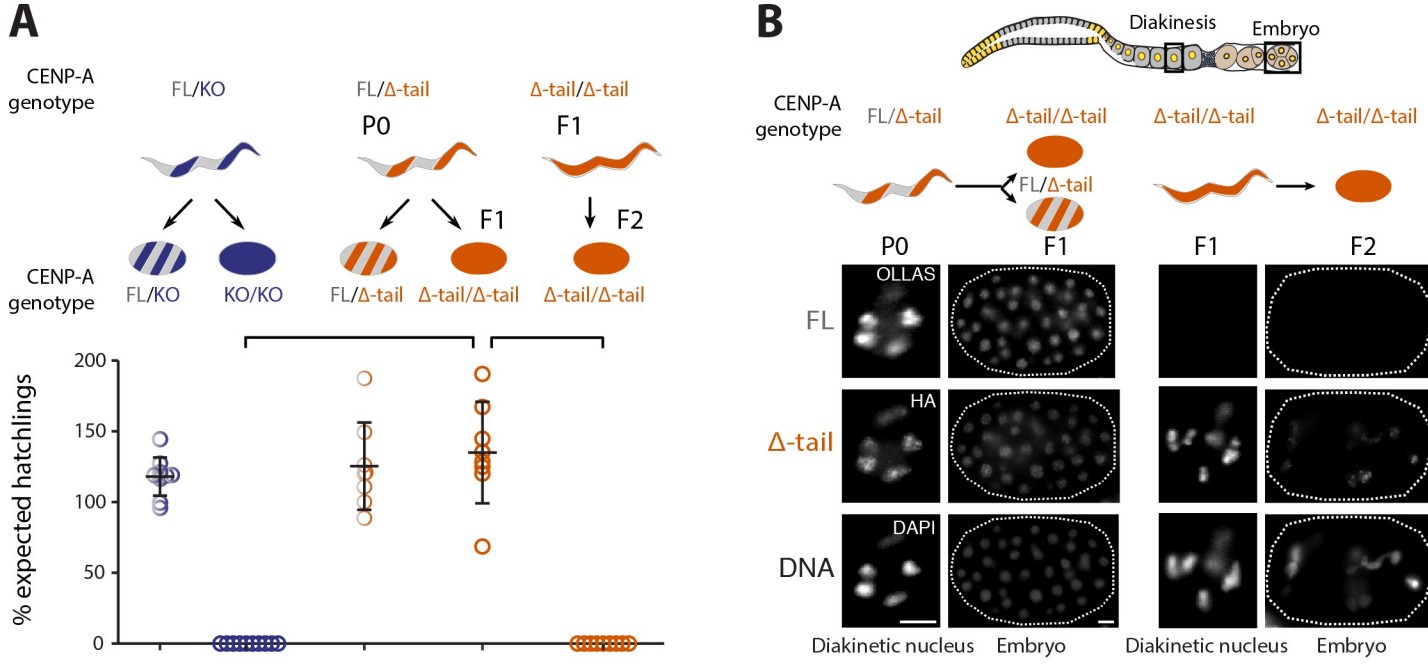

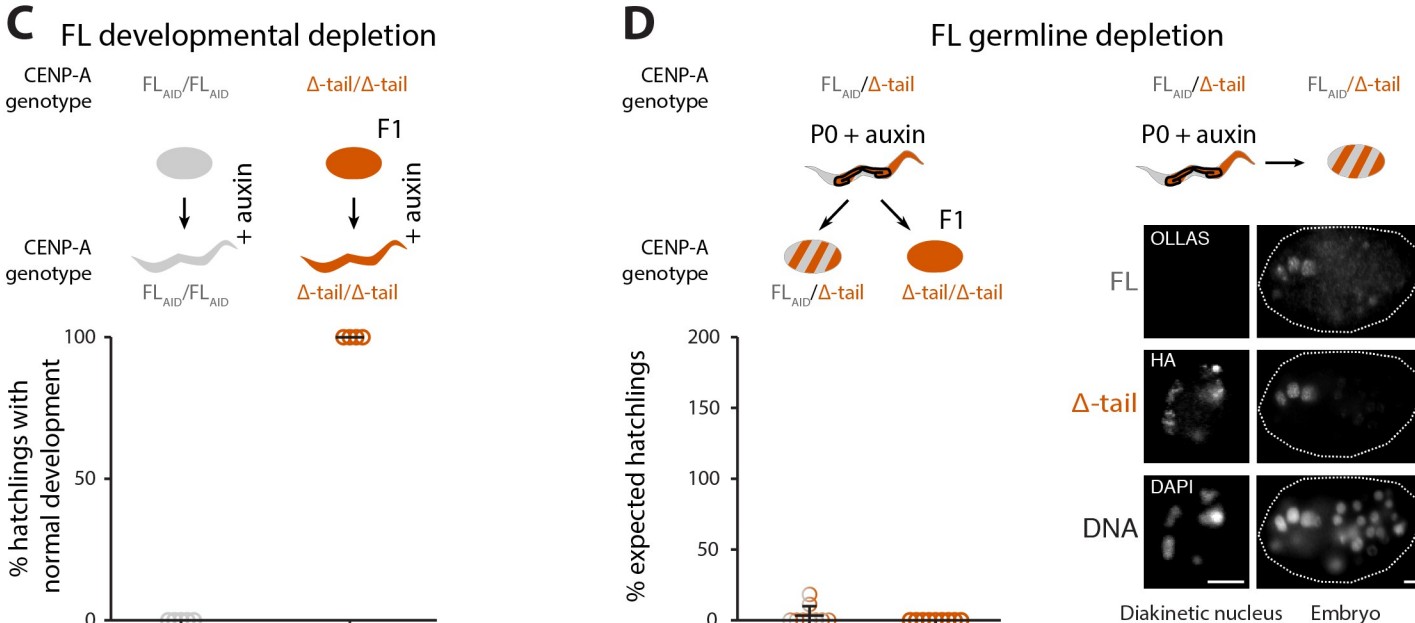

**Fig 3. The presence of the CENP-A N-terminal tail is dispensable for mitosis during development but required in the germ line.** (A, B) Phenotypic consequences of the CENP-A N-terminal tail deletion in presence or absence of CENP-A FL in the maternal germ line. (A) Quantification of viable offspring of worms carrying CENP-A FL (gray), CENP-A KO mutation (dark blue), or CENP-A tail truncation (Δ-tail, orange) alleles. A balancer allele (hT2) was used to maintain heterozygous strains. In these balancer strains, 10/16 of the offspring are aneuploid, and CENP-A FL homozygotes (1/16) are embryonic lethal due to the balancer alleles and are therefore not shown. Top, cartoon of maternal heterozygous (striped) or homozygous (full) genotypes as well as genotypes of the offspring. Bottom, quantification of the viable offspring as a percentage of the expected hatchlings in the context of the balancer allele. Error bars show the standard deviation of the mean. Connecting lines highlight the most relevant comparisons between the different genotypes. $N$ = 8 to 10 independent broods. (B) Cartoon of an adult germ line, with imaged regions highlighted with black boxes. IF images of both the CENP-A FL and CENP-A Δ-tail proteins in the maternal germ line (diakinesis) and embryos. Left, maternal CENP-A FL/Δ-tail

P0 giving rise to either CENP-A FL/Δ-tail or Δ-tail/Δ-tail F1 offspring (indistinguishable as early embryos). Right, maternal CENP-A Δ-tail/Δ-tail F1 giving rise to CENP-A Δ-tail/Δ-tail F2 offspring. Cartoon images with genotype color codes as in (A). (C) Depletion of the CENP-A FL protein during development in CENP-A FL and Δ-tail (F1) homozygous worms, using the AID system. Embryos were placed onto auxin-containing plates for depletion during development. Top, cartoon of the genotypes with color codes as in (A). Bottom, quantification of hatchlings with normal development upon CENP-A FL protein depletion. $N = 9$ independent broods. (D) Depletion of the CENP-A FL protein in the germ line of CENP-A FL/Δ-tail heterozygous worms (P0), using the AID system. Adults were placed on auxin plates for 4 h. Left, cartoon of maternal genotype and genotypes of the offspring, with color codes as in (A), and quantification of the viable offspring as a percentage of the expected hatchlings in the context of the balanced allele. $N = 4$–5 independent broods. Right, IF images of both the CENP-A FL and CENP-A Δ-tail proteins in the maternal germ line (diakinesis) and embryos. Adult CENP-A FL/Δ-tail P0 upon AID-mediated depletion of CENP-A FL protein in the germ line, giving rise to CENP-A FL/Δ-tail offspring. Scale bars in (B, D) represent 5 μm. The data underlying all the graphs can be found in S1 Data. AID, auxin-inducible degron; FL, full-length; IF, immunofluorescence; KO, knockout.

OLLAS-tagged CENP-A FL and either CENP-A Δ-tail or CENP-A KO. We again found that the maternally contributed CENP-A FL protein became undetectable mid-gastrulation (S2B Fig), and the maternal contribution is thus independent of the genotype of the embryos (FL/FL, Δ-tail/Δ-tail, or KO/KO; S2C Fig). Homozygous CENP-A KO embryos arrest when maternal contribution runs out, but homozygous CENP-A Δ-tail F1 embryos continue to develop into adults. Consistent with maternal CENP-A FL protein only being detectable in early embryos, we did not detect any CENP-A FL protein in the germ line of adult CENP-A Δ-tail F1 worms by immunofluorescence (IF) (Fig 3B). These results demonstrate that after the maternally provided CENP-A FL protein runs out (around 100 to 200 cell stage), CENP-A Δ-tail alone is functional in mitosis and able to support late embryonic and larval development and germline proliferation.

To exclude the possibility that CENP-A Δ-tail protein in the CENP-A Δ-tail homozygous F1 stabilizes undetectable amounts of CENP-A FL protein that could support mitosis into adulthood, we depleted CENP-A FL protein during development. We used the auxin-inducible degron (AID) system [36] and fused CENP-A FL with a degron tag. We then combined it with a soma-specific TIR1 driver that is expressed in most or all somatic tissues throughout development, from embryos to adults [36]. We then placed embryos onto auxin-containing plates for development. In homozygous CENP-A FL animals, this led to embryonic or larval lethality (Fig 3C). In contrast, homozygous CENP-A Δ-tail F1 embryos hatched and developed normally into adult worms (Fig 3C). Although we already did not detect CENP-A FL protein in these CENP-A Δ-tail F1 animals before depletion (Fig 3B), this experiment demonstrates that there is no pool of undetectable CENP-A FL protein that persists through development and that CENP-A Δ-tail protein is capable of supporting mitosis in the absence of the CENP-A FL protein.

Together, these results show that the presence of the CENP-A N-terminal tail is dispensable for mitosis during development and germline proliferation.

## Full-length CENP-A is required for de novo centromere establishment in the maternal germ line

The experiments described above demonstrate that CENP-A Δ-tail is able to support chromosome segregation during mitosis and that functional centromeres can be maintained throughout development in absence of the CENP-A N-terminal tail. They also narrow down the window in which the presence of CENP-A FL is essential from diplotene of the maternal germ line (where it first reappears; Fig 1) to the approximately 200 cell stage embryos (where it disappears; S2 Fig) and leave 2 possible hypotheses: (1) Mitosis in early embryogenesis is functionally distinct from mitosis post-gastrulation, and the former requires the N-terminal tail of CENP-A, whereas the latter does not. (2) The N-terminal tail is required before the first embryonic mitosis in the maternal germ line.

It is difficult to envision why the CENP-A N-terminal tail would be essential for chromosome segregation in early, but not in late embryos, and we therefore considered how centromeres are established in the germ line and inherited into the embryos. The F1 embryos inherit their centromeres from heterozygous CENP-A FL/Δ-tail P0 germ lines, where CENP-A FL is present during the centromere establishment step in prophase of meiosis I. The F2 embryos, however, inherit their centromeres from homozygous CENP-A Δ-tail F1 germ lines, where no CENP-A FL is present. We therefore hypothesized that the CENP-A N-terminal tail is essential for centromere de novo establishment in the maternal germ line or during the maternal-to-zygotic transition. If this is the case, depletion of the CENP-A FL protein from the germ line of heterozygous CENP-A FL/Δ-tail P0 animals should result in embryonic lethality of all F1 offspring, including the ones homozygous for the CENP-A Δ-tail. To test this, we depleted the CENP-A FL protein in adult heterozygous CENP-A FL/Δ-tail P0 worms by using AID with a germline-specific TIR1 driver. This treatment removed all detectable CENP-A FL protein from the germ line, while still allowing zygotic expression in the embryo (Fig 3D). Upon CENP-A FL protein depletion in the P0 germ line, we observed severe chromosome missegregation in all F1 embryos, which mimics the *emb* phenotype observed in the F2 homozygous generation (Fig 3D).

If the presence of the CENP-A N-terminal tail is required only for centromere establishment, but not for development, it should be possible to maintain homozygous CENP-A Δ-tail animals for generations as long as some CENP-A FL protein is present in the germ line during the centromere establishment phase in each generation. To test this, we crossed CENP-A Δ-tail homozygous males to heterozygous CENP-A FL/Δ-tail P0 hermaphrodites. With the resulting homozygous CENP-A Δ-tail males, the cross can be repeated for at least 7 generations, with the males never expressing CENP-A FL protein from their own genome (Fig 4A). They are nevertheless viable, as CENP-A FL is present in the maternal germ line where the germ cells that will give rise to the males mature, and where their centromeres are established.

These results predict that although homozygous CENP-A KO embryos and homozygous CENP-A Δ-tail F2 embryos are both inviable, the cause of the embryonic lethality is different: The former inherit correctly established centromeres but die because they cannot zygotically express CENP-A protein to maintain mitotic segregations, while the latter die because the centromere establishment defects in the maternal germ line result in early embryonic chromosome segregation defects. Consistent with this interpretation, the CENP-A KO embryos can be rescued by zygotic expression of either CENP-A FL or CENP-A Δ-tail protein, which is introduced by crossing CENP-A FL or Δ-tail homozygous males to CENP-A FL/KO heterozygote hermaphrodites (Fig 4B). In contrast, the embryonic lethality of CENP-A Δ-tail F2 embryos cannot be rescued by zygotic expression of CENP-A FL protein (Fig 4C).

Our results are consistent with a role of the CENP-A N-terminal tail in centromere establishment, but the precise developmental window in which it is required is difficult to determine because of the maternal contribution of the protein into the early embryos. To narrow down the essential developmental window for the CENP-A N-terminal tail, we expressed a pulse of CENP-A FL in adult CENP-A Δ-tail F1 homozygotes from a transgene under the control of a heat shock promoter. We then examined embryonic viability of the F2 embryos 0 to 8 h after the heat shock. Expression of CENP-A FL did not rescue the embryos produced in the first 2 h after the heat shock, despite the protein already being present (Figs 4D and S3). However, embryos produced 3 to 5 h after the heat shock were viable and showed wild-type chromosome segregation, with almost 100% viability at 4 h (Fig 4D). At later time points, viability returned to 0%, as the CENP-A FL protein produced during the heat shock ran out (Figs 4D and S3). The delay in rescue strongly suggests that the CENP-A FL is required in the germ line, prior to the onset of the first embryonic cell divisions, and that only germ cells that are in

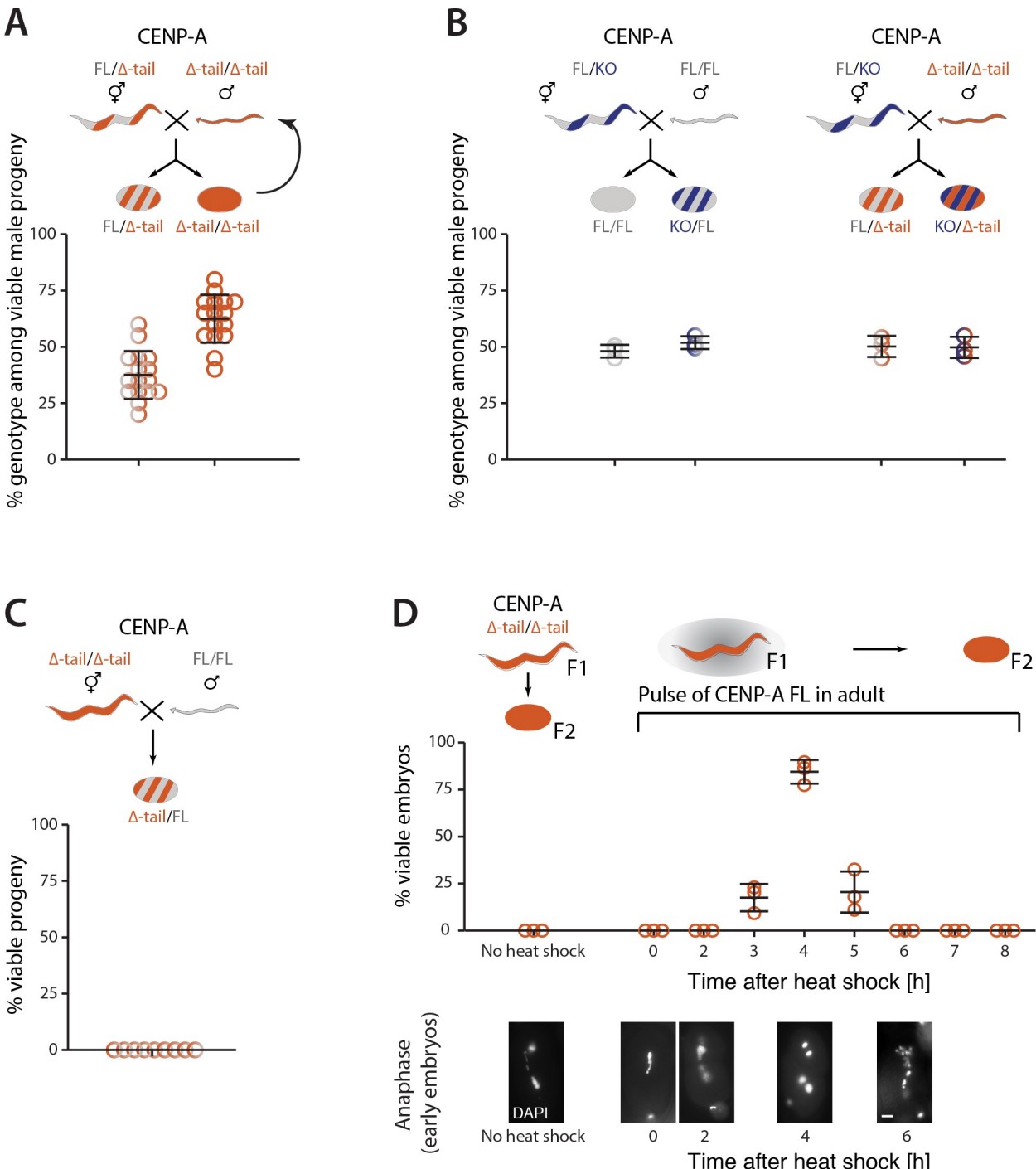

**Fig 4. The CENP-A N-terminal tail is required for establishing centromere identity in the germ line.** Cartoons showing the different CENP-A genotypes (FL, gray; KO, dark blue; Δ-tail, orange) as heterozygotes (striped) or homozygotes (full). (A) Maintenance of CENP-A Δ-tail homozygous males by crossing. Percentage CENP-A Δ-tail hetero- and homozygotes among viable male cross-progeny after crossing CENP-A Δ-tail homozygous males with heterozygous CENP-A FL/Δ-tail hermaphrodites. $N$ = 16 independent crosses. (B) Rescue of embryonic lethality of CENP-A KO embryos by zygotic expression of CENP-A FL (left) or Δ-tail (right), introduced by genetic crosses of CENP-A FL or Δ-tail homozygous males to heterozygous CENP-A KO hermaphrodites. Percentages of viable progeny for each genotype are shown. $N$ = 3 independent crosses each. (C) Non-rescue of embryonic lethality of CENP-A Δ-tail F2 embryos by zygotic expression of CENP-A FL, introduced by genetic crossing of wild-type males to homozygous CENP-A Δ-tail F1 hermaphrodites. Percentage of viable progeny is shown. $N$ = 9 independent crosses. (D) Rescue of the embryonic lethality of CENP-A Δ-tail F2 embryos by pulse of CENP-A FL protein expression in F1 adults. (D) Percentage of viable F2 embryos without heat shock and at different time points 0–8 h after heat shock–induced expression of CENP-A FL protein in adult F1

(top). Images of DAPI-stained 1- or 2-cell F2 embryos during anaphase at the indicated time points after heat shock (bottom). *N* = 3 independent heat shock experiments. Error bars show the standard deviation of the mean. The data underlying all the graphs can be found in S1 Data. CENP-A, centromere protein A; FL, full-length; KO, knockout.

a specific stage during the pulse of heat shock–induced CENP-A FL expression can later give rise to viable embryos. We determined that during the 4 h after the heat shock, each gonadal arm produces about 15 ± 3 embryos, which puts the precursors of the viable embryos into diplotene stage at the time of the heat shock, and corresponds to the stage when CENP-A is reestablished in the germ line.

These results support our hypothesis that the CENP-A N-terminal tail is required for centromere establishment in the proximal germ line and that once centromeres are established, CENP-A Δ-tail is able to maintain centromeres and support mitosis during development.

## CENP-A Δ-tail F2 embryos show aberrant centromeres and a kinetochore-null phenotype

We next investigated the centromere formation defects in the F2 embryos in more detail. We decided to use the striking biorientation of CENP-A and KNL-2 on the metaphase plate in mitotic cells (both in the distal mitotic zone of the germ line and in embryos) as a visible readout of a correctly assembled centromere. As expected, mitotic cells in the germ line of P0 CENP-A FL/Δ-tail heterozygotes show this biorientation (Fig 5A). In the F1 CENP-A Δ-tail homozygote generation, biorientation is visible in embryonic mitotic cells and in adult mitotic germ cells (Fig 5A). This observation supports the model that once centromere identity is established in the heterozygous germ line of the P0, it can be inherited into the adult germ line of the homozygous F1 (Fig 5A). In F2 embryonic mitotic cells, localization of both CENP-A Δ-tail and KNL-2 becomes disordered (Fig 5A). CENP-A Δ-tail still appears on chromatin but is no longer bioriented. KNL-2 hardly remains present on chromatin and instead clusters in large dots that colocalize with the microtubules (Figs 5A and S4; S1 and S2 Videos). Colocalization of KNL-2 with microtubules upon loss of CENP-A was already observed in an earlier study [30]. F2 embryos also show chromosomes that are less compacted (Fig 5A), which is similar to the condensation defects observed upon loss of CENP-A or KNL-2 [30].

The mislocalization of both CENP-A Δ-tail and KNL-2 in the first embryonic cell division of F2 CENP-A Δ-tail homozygous embryos leads to a dramatic missegregation of chromosomes, which explains the penetrant *emb* phenotype (Figs 3A and 5A; S3 and S4 Videos). Live imaging of KNL-2 and the kinetochore components HCP-4 and ROD-1 showed that no kinetochore is formed during the first embryonic cell division in F2 embryos (Fig 5B, S5–S8 Videos). This *knl* phenotype is also apparent when looking at the speed at which the spindle poles move away from each other following nuclear envelope breakdown (NEBD) (Fig 5C and 5D; S3 and S4 Videos). As no physical connection can be established between the chromosomes and microtubules, the lack of tension causes the spindle poles to move away from each other prematurely.

Since CENP-A Δ-tail is able to support functional mitosis, as observed in F1 animals during development, it is difficult to directly explain the *knl* phenotype of F2 embryos by the lack of the CENP-A N-terminal tail in these cells. Instead, we propose that it is caused by the failure in establishing centromere identity in absence of the CENP-A N-terminal tail in the germ line of F1 CENP-A Δ-tail homozygotes.

CENP-A has previously been shown to be dispensable for chromosome segregation during meiosis in *C. elegans* [27]. We therefore considered it unlikely that the observed *knl* phenotype in the first mitotic division of F2 CENP-A Δ-tail homozygous embryos is the result of meiotic

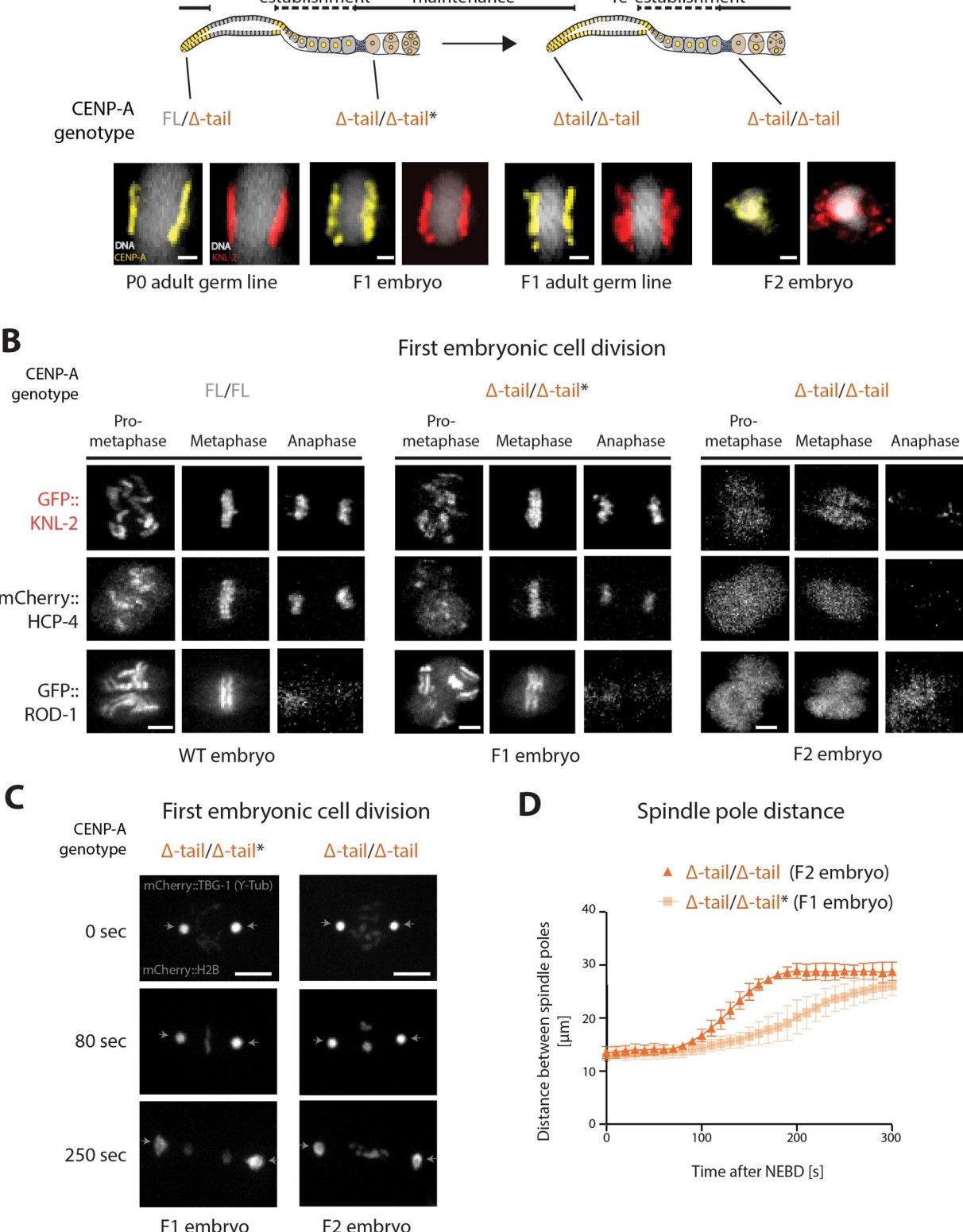

**Fig 5. Centromere identity and function are maintained for one generation upon deletion of the CENP-A N-terminal tail.** (A) Centromere biorientation in metaphase cells. IF images of KNL-2 and CENP-A Δ-tail in metaphase cells in the mitotic zone of the adult hermaphrodite germ line or in early embryos for the different generations (P0, F1, F2) of the CENP-A Δ-tail strain. (B) Still images for the indicated stages of mitosis

of recordings in embryos derived from CENP-A FL/FL, FL/Δ-tail, or Δ-tail/Δ-tail maternal germ lines. KNL-2 (GFP), HCP-4 (mCherry), or ROD-1 (GFP) were fluorescently labeled. (C) Still images of recordings in 1-cell embryos derived from CENP-A FL/Δ-tail or Δ-tail/Δ-tail maternal germ lines, showing GFP-histone H2B (chromosomes) and GFP–γ-tubulin (spindle poles). Time is given in seconds relative to NEBD. The arrows indicate the location of the spindle poles. (D) Quantification of spindle pole separation kinetics during the first embryonic cell division in embryos derived from CENP-A FL/Δ-tail or Δ-tail/Δ-tail maternal germ lines. Error bars show the standard deviation of the mean. Scale bars represent 1 μm (A), 4 μm (B), or 10 μm (C). CENP-A Δ-tail homozygote F1 embryos are indistinguishable from heterozygous embryos; therefore, the F1 genotypes are marked with an asterisk in all panels. The data underlying all the graphs can be found in S1 Data. CENP-A, centromere protein A; FL, full-length; IF, immunofluorescence; NEBD, nuclear envelope breakdown; WT, wild-type.

chromosome segregation defects. Indeed, the meiotic divisions, which happen in the fertilized embryo prior to the onset of the mitotic divisions, appear unaffected (S4B Fig). Therefore, the first instance of observable chromosome segregation defects upon deletion of the CENP-A N-terminal tail appears during the first F2 embryonic division.

Nevertheless, some altered localization patterns of CENP-A Δ-tail and KNL-2 are already observable in the F1 CENP-A Δ-tail animals, before the chromosome segregation defects become apparent in the F2 embryos (S5 Fig). In the mitotic zone of the germ line of F1 animals, both CENP-A Δ-tail and KNL-2 show a punctate pattern. KNL-2 foci are also visible in the proximal germ line of F1 animals (S5 Fig). Moreover, the removal of both CENP-A and KNL-2 at the mitosis-to-meiosis transition is delayed in some of the F1 animals, where both proteins can be detected in part of the pachytene region of the germ line (S5 Fig). Although these changes in localization and distribution of CENP-A Δ-tail and KNL-2 are visible in distal germ cells (S5 Fig), they do not explain the centromere defects that lead to the *knl* phenotype observed in the F2 embryos. Despite the punctate patterns of CENP-A Δ-tail and KNL-2, mitosis appears normal in these cells, based on CENP-A Δ-tail and KNL-2 biorientation at metaphase (Fig 5A) and the fact that F1 animals show normal germline proliferation. The implications of the CENP-A Δ-tail pachytene presence are also not clear, since it is visible in only some of the F1 animals, whereas the F2 embryonic lethality is fully penetrant (Fig 3A; S5 Fig). However, in the diplotene and diakinesis regions, where CENP-A reappears on chromatin and where centromere identity is established, the chromosomes are very compact, potentially obscuring differences between wild type, P0, and F1. We therefore next considered how CENP-A is loaded onto chromatin.

## De novo loading of full-length CENP-A in the *C. elegans* proximal germ line requires KNL-2

CENP-A and KNL-2 have previously been shown to be codependent for chromatin association in embryonic cells [30]. To test the interdependence of CENP-A FL and KNL-2 in the germ line, we acutely removed these proteins from chromatin by using the AID system with a germline-expressed TIR1. Depletion of KNL-2 led to the expected co-depletion of CENP-A FL in the proximal germ line (Fig 6A). However, depleting CENP-A FL did not lead to loss of KNL-2 in the proximal germ line, as was reported in embryos [30]. Instead, KNL-2 remained largely associated with chromatin (Fig 6A). The persistence of KNL-2 in proximal germ cells upon depletion of CENP-A FL suggests that KNL-2 acts upstream of CENP-A in the de novo establishment of centromeres, which may explain the essential role of the CENP-A N-terminal tail in this process.

Despite the reliance on the CENP-A N-terminal tail for the interaction with KNL-2, CENP-A Δ-tail is visible on chromatin in F1 CENP-A Δ-tail homozygous germ lines (Figs 5A and 6B). However, in contrast to CENP-A FL, it remains on chromatin upon depletion of KNL-2 in the germ line of F1 animals (Fig 6B). This shows that the chromatin association of

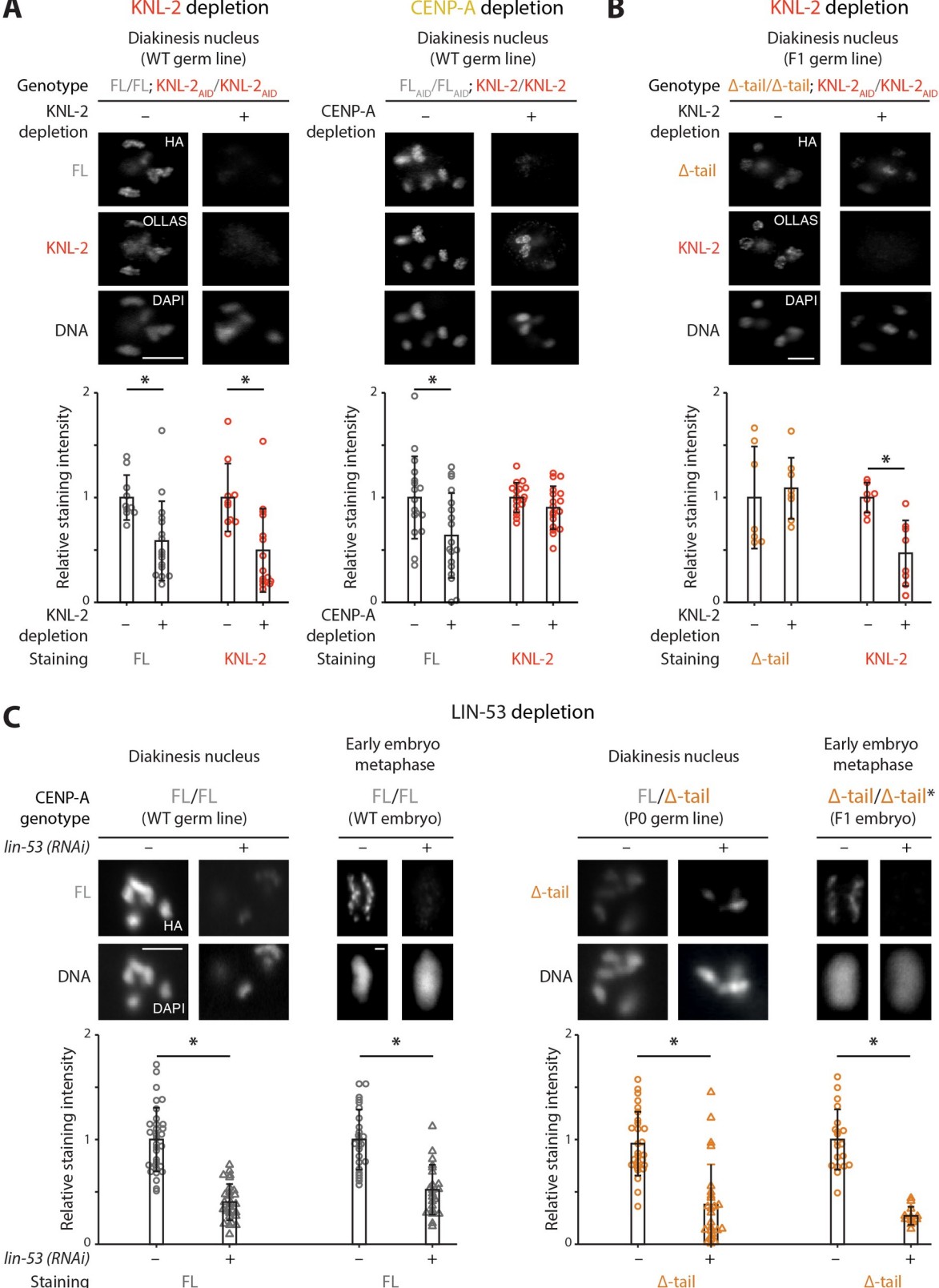

**Fig 6. CENP-A Δ-tail is maintained on chromatin by LIN-53 in absence of the interaction with KNL-2.** (A) Germline-specific depletion of CENP-A FL and KNL-2 using the AID system. IF images of CENP-A FL and KNL-2 in diakinesis nuclei of adult

hermaphrodite germ lines, with and without auxin-induced degradation of either KNL-2 (left) or CENP-A FL (right). Bar plots show quantifications of total nuclear fluorescence in diakinesis nuclei. $N$ = 10–18 nuclei total, from 3 independent IF experiments. (B) Chromatin association of CENP-A Δ-tail in presence or absence of KNL-2. IF images and fluorescence quantifications of CENP-A Δ-tail and KNL-2 in diakinesis nuclei of CENP-A Δ-tail F1 homozygous adult hermaphrodite germ lines. KNL-2 was depleted using germline-specific AID. N = 7–8 nuclei total, from one IF experiment. Germline-specific depletion of KNL-2 and CENP-A in (A) and (B) was achieved using a TIR1 with germline- and early embryo-restricted expression. (C) Chromatin association of CENP-A FL and Δ-tail in presence or absence of LIN-53. IF images showing CENP-A FL and CENP-A Δ-tail in diakinesis nuclei and at metaphase in early embryos, for CENP-A FL homozygotes (left) and CENP-A FL/Δ-tail heterozygote P0 adults and CENP-A Δ-tail homozygote F1 embryos (right) (indistinguishable from heterozygous embryos; therefore, the F1 genotype is marked with an asterisk). LIN-53 was depleted by RNAi. Bar plots show quantifications of CENP-A levels in control and LIN-53-depleted animals. $N$ = 26–33 oocytes and 12–26 embryonic metaphase plates from 3 independent IF experiments. In all quantifications, the mean level of the control (no auxin or no RNAi) in each experiment was set to 1. Error bars show the standard deviation of the mean, and the asterisks denote statistical significance, determined by using a Student $t$ test. Scale bars represent 5 μm (diakinesis oocytes) or 1 μm (metaphases). The data underlying all the graphs can be found in S1 Data. AID, auxin-inducible degron; CENP-A, centromere protein A; FL, full-length; IF, immunofluorescence; RNAi, RNA interference; WT, wild-type.

CENP-A Δ-tail does not depend on KNL-2, which may be expected given the loss of interaction between these two proteins (Fig 2).

To assess additional differences, we quantified the levels of CENP-A FL and Δ-tail in heterozygous and homozygous diakinesis oocytes and early embryonic metaphases. We found that levels of CENP-A Δ-tail were consistently lower compared to CENP-A FL (S6 Fig), and the effect was stronger for homozygous cells (S6A and S6B Fig) compared to heterozygous cells (S6C and S6D Fig). These results suggest that CENP-A Δ-tail is either differently and less efficiently assembled onto chromatin or that CENP-A Δ-tail-containing nucleosomes are less stable. We note, however, that despite the reduced levels of CENP-A Δ-tail in F1 animals, their development appears completely normal.

Taken together, we find that KNL-2 acts upstream of CENP-A FL deposition and that CENP-A Δ-tail deposition does not depend on KNL-2. We therefore hypothesize that the genomic incorporation of CENP-A Δ-tail in the F1 proximal germ line differs from that of CENP-A FL in wild-type proximal germ lines.

## CENP-A Δ-tail is maintained on chromatin by LIN-53

The reliance on the CENP-A N-terminal tail for the interaction with KNL-2 (Fig 2) and the chromatin association of CENP-A Δ-tail in absence of KNL-2 (Fig 6B) imply that the maintenance of CENP-A Δ-tail during development of the F1 CENP-A Δ-tail homozygous animals is mediated by factors other than KNL-2. The RbAp46/48 homolog LIN-53 has been shown to be important for CENP-A chromatin association in *C. elegans* [31]. We found that depleting LIN-53 by RNA interference (RNAi) resulted in reduced levels of both CENP-A FL and CENP-A Δ-tail, both in the proximal germ line and in embryos (Fig 6C). These results confirm the role of LIN-53 in maintaining CENP-A on chromatin and suggest that LIN-53 is involved in the observed centromere maintenance in the homozygous CENP-A Δ-tail F1 animals during development. As depletion of LIN-53 affects levels of both CENP-A FL and Δ-tail (Fig 6C), LIN-53 may also be involved in the initial centromere establishment in the proximal germ line. Depletion of LIN-53 by RNAi results in comparable embryonic lethality among the offspring of both wild-type and P0 CENP-A Δ-tail heterozygous animals (S7 Fig), consistent with what has previously been observed [31].

In summary, we propose a functional distinction between centromere establishment and centromere maintenance in *C. elegans*. Although both require CENP-A, the CENP-A N-terminal tail is specifically required to reestablish correct centromere identity in the proximal adult hermaphrodite germ line in every generation and mediates an interaction with KNL-2. Once established, this centromere identity can then be inherited and functional centromeres

maintained throughout development in the absence of the CENP-A N-terminal tail, but only until the next generation, where resetting of centromere identity fails (Fig 7A).

## Discussion

In this study, we take advantage of the CENP-A dynamics in the *C. elegans* adult hermaphrodite germ line to investigate the regulation of centromere formation and inheritance. In most eukaryotes, CENP-A is maintained by an epigenetic mechanism from one cell cycle to the next. DNA replication dilutes CENP-A nucleosomes at established functional centromeres, and the remaining CENP-A nucleosomes act as epigenetic marks for the reloading of new CENP-A at the same genomic regions at cell cycle stages that vary between organisms [10,37–39]. Centromeres are also maintained from one generation to the next, which has been studied in a number of model organisms. CENP-A molecules in mouse oocytes show remarkable stability during mouse meiosis [25]. In contrast, CENP-A in starfish oocytes is actively replenished at centromeres [24]. This mechanism is dependent on the MIS18 complex, as well as the chaperone HJURP, and active transcription to remove the old CENP-A–containing nucleosome.

In *C. elegans*, the cycle of inheritance is interrupted at the onset of meiosis in hermaphrodites, and centromeres have to be reestablished during diplotene/diakinesis of meiosis I in the proximal germ line (Fig 1). How centromeres are established de novo is not well understood, and most studies so far have relied on the observation of spontaneous or artificially induced vertebrate neocentromeres or artificial chromosomes. The factors required for this process are similar to those required for centromere maintenance. HJURP, CENP-C, and CENP-I have all been implicated in centromere establishment by studying the formation of neocentromeres in human and chicken cell lines [40–42]. The MIS18 complex or M18BP1/KNL-2 has not been reported to be sufficient for neocentromere establishment, as this complex is believed to only recognize existing centromeres through interactions with CENP-C or CENP-A [43,44]. However, nonmammalian homologs of KNL-2 often show cell cycle dynamics that are different from the human homolog, and localize to centromeres throughout the cell cycle, suggesting that they may have taken on additional roles [30,43–46]. Consistently, we find in this study that *C. elegans* KNL-2 is involved in both centromere establishment and maintenance. However, its role may be different during the two processes. CENP-A FL and KNL-2 are codependent for chromatin association during maintenance, as previously shown [30], but CENP-A Δ-tail remains chromatin bound (Fig 6B) and is fully functional in centromere maintenance (Fig 3), even though its interaction with KNL-2 is impaired (Fig 2). During centromere establishment, the interaction of KNL-2 and CENP-A is essential, and we find that KNL-2 may act upstream of CENP-A during this process (Fig 6A).

Our results show that the interaction between KNL-2 and CENP-A depends on the central region of KNL-2 and the N-terminal tail of CENP-A in *C. elegans* (Fig 2). These findings have recently been independently confirmed [34]. In chicken cells, the interaction between KNL-2 and CENP-A was shown to be dependent on a KNL-2 CENP-C motif [44]. Such a CENP-C motif has been identified in a wide variety of taxa, including fish, frogs, and plants [45,47,48]. In *A. thaliana*, disrupting the CENP-C motif leads to a failure of KNL-2 to bind to centromeric chromatin [47]. Additionally, a detailed study of *Xenopus laevis* KNL-2 shows that the CENP-C motif regulates interaction with CENP-A nucleosomes [45]. Considering the conserved nature of the CENP-C motif, it is perhaps surprising that it cannot be found in *C. elegans* KNL-2 [49]. Instead, the minimal region for interaction with CENP-A FL in Y2H had already been shown to be a larger central region of KNL-2 (aa 269–470) [33]. We could

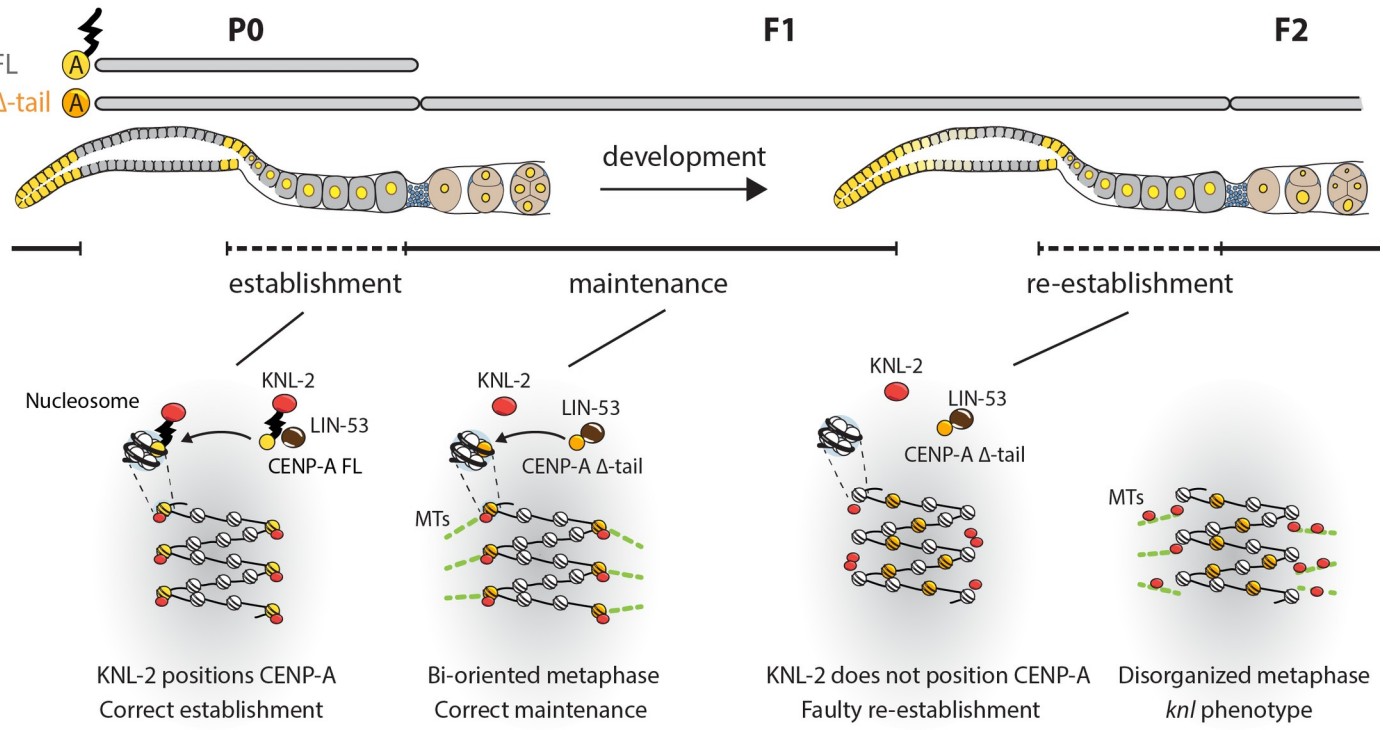

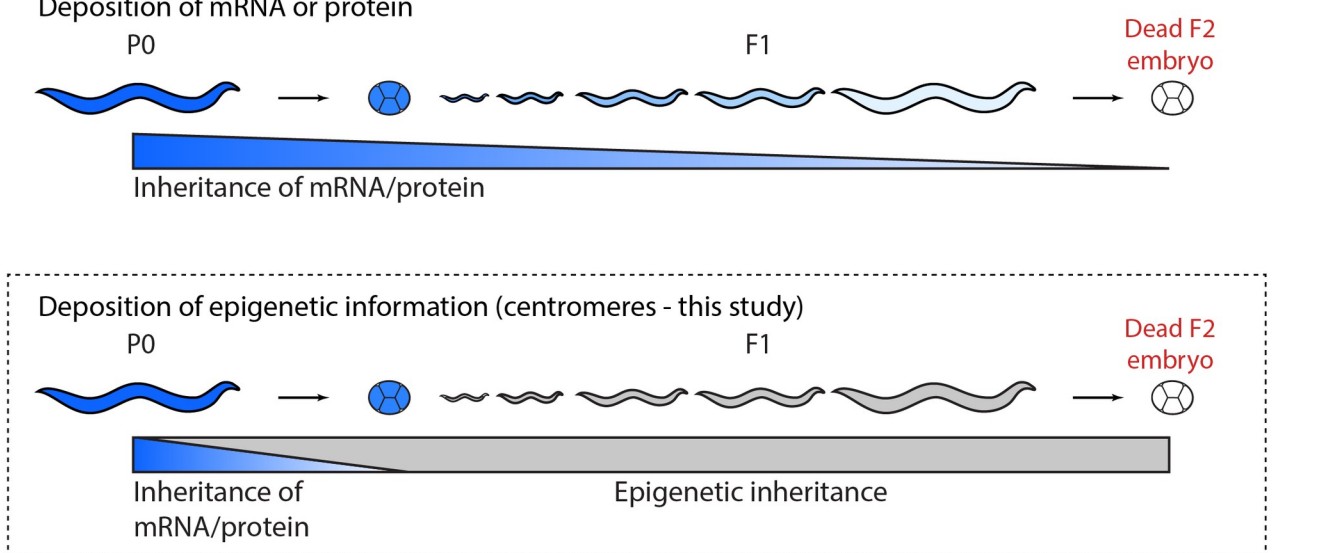

**Fig 7. Model for the establishment and maintenance of centromere identity and the role of the CENP-A N-terminal tail.** (A) Model of centromere establishment and maintenance across generations in *C. elegans*. Centromeres are established in the proximal germ line and maintained until their removal in the distal germ line of the next generation. The CENP-A N-terminal tail interacts with KNL-2 and is required for establishing centromeres. The presence of CENP-A on

chromatin also depends on LIN-53. In the heterozygous P0 worm, the CENP-A FL present in the germ line is sufficient for correct centromere establishment. Upon loss of the CENP-A N-terminal tail, centromeres can be maintained for one generation, a process that is likely mediated by LIN-53. However, in absence of the CENP-A N-terminal tail, centromere establishment fails in the adult F1 generation, leading to mislocalization of CENP-A Δ-tail and KNL-2, and a *knl* phenotype in the F2 embryos. (B) Two models explaining a *mel* phenotype, where the function of an essential gene is maintained for one generation despite loss or truncation of the gene. Top, sufficient protein or mRNA is deposited in the embryo from the maternal germ line for it to persist and function during development. Bottom, the protein sets a (chromatin) state in the parental germ line that serves as a memory that persists during development by epigenetic inheritance. The centromere inheritance described in this study follows the epigenetic inheritance model, as indicated by the dotted line. Protein presence is labeled in blue, and epigenetic inheritance in gray. CENP-A, centromere protein A; FL, full-length; *knl*, kinetochore-null; *mel*, maternal effect lethal; mRNA, messenger RNA; MTs, microtubules.

confirm this result and demonstrate that the interaction is mediated by the N-terminal tail of CENP-A (Figs 2 and S1A).

The N-terminal tail of CENP-A is considered to be flexible and unstructured but plays an important role in centromere biology [50–54]. It contributes to the interactions with the kinetochore proteins CENP-B, CENP-C, and CENP-T [50,51,55,56], although there may be species-specific differences in how these proteins interact. Different posttranslational modifications (PTMs) on the CENP-A N-terminal tail have also been described to affect centromere establishment and maintenance [52,53,57–62]. Here, we describe a role for the CENP-A N-terminal tail in establishing centromere identity in the hermaphrodite germ line of *C. elegans*. In *A. thaliana*, the CENP-A N-terminal tail is required for meiosis, and CENP-A N-terminal tail replacements caused defects in meiotic CENP-A loading, defects in meiotic segregation, and led to sterility [63,64]. Although removing the CENP-A N-terminal tail in *C. elegans* does not affect meiotic chromosome segregation (S4B Fig), we do find that it has a role during earlier stages of meiosis, where it is required to set centromere identity for the next generation. In fission yeast, removing the CENP-A N-terminal tail does not affect the loading of CENP-A at centromeres, but the N-terminal tail confers centromere stability in mitotically dividing cells, most likely through an epigenetic process [56]. In mammalian cells, the CENP-A N-terminal tail has also been shown to be dispensable for cell viability, but it works in concert with the centromere targeting domain (CATD) to recruit CENP-C and CENP-T [50,51]. The distinct requirements for the CENP-A N-terminal tail in centromere establishment and centromere maintenance are therefore likely not unique to *C. elegans*.

We report that CENP-A lacking the N-terminal tail is no longer dependent on KNL-2 for chromatin association (Fig 6B). Strikingly, CENP-A Δ-tail is still functional in mitotic chromosome segregation (Figs 3 and 4), suggesting that it is sufficient for centromere maintenance and that KNL-2 has essential functions in mitosis other than CENP-A loading. LIN-53, a homolog of the histone chaperone RbAp46/48, has been implicated in maintaining CENP-A chromatin association in *C. elegans* [31]. We find that this is the case also for CENP-A Δ-tail, both in the germ line and in embryos (Fig 6). As depletion of LIN-53 affects both CENP-A FL and Δ-tail, LIN-53 may also be involved in centromere establishment, which would be consistent with the recently reported role of LIN-53 in de novo centromere establishment on *C. elegans* artificial chromosomes [65].

We find that the interaction between KNL-2 and the CENP-A N-terminal tail appears essential during centromere establishment before the onset of the first embryonic cell division, and only KNL-2 assembled CENP-A seems to be competent in setting centromere identity. We propose that the interaction of CENP-A with KNL-2 through the CENP-A N-terminal tail is required to target CENP-A to the correct genomic regions to establish proper centromere identity in the germ line. Once established, this centromere identity can then be maintained in the absence of the CENP-A N-terminal tail for one generation.

LIN-53 could act in this epigenetic centromere maintenance by targeting CENP-A to the centromeric locations predefined in the maternal germ line. We still observe the LIN-53–

dependent chromatin association of CENP-A Δ-tail in F1 CENP-A Δ-tail homozygotes, but we speculate that this chromatin association has an altered genomic distribution. LIN-53 could normally work in concert with KNL-2 to establish functional centromeres, but in the absence of CENP-A Δ-tail interaction with KNL-2 in the F1 germ line, LIN-53 could establish CENP-A Δ-tail–containing chromatin in random genomic regions, resulting in a failure to set correct centromere identity. LIN-53 would subsequently maintain the random CENP-A Δ-tail incorporation patterns, resulting in the lack of CENP-A Δ-tail and KNL-2 biorientation and the loss of centromere function in the F2 embryos. This model (Fig 7A) suggests that even in the context of holocentricity, correct genomic CENP-A localization is important for centromere function.

While the CENP-A Δ-tail homozygous F1 generation is viable, and CENP-A Δ-tail is sufficient to properly assemble the kinetochore, the CENP-A Δ-tail F2 embryos show the characteristic features of a *knl* phenotype (Fig 5) [32,66,67]. F1 and F2 homozygous embryos are genetically identical, and we therefore propose that the kinetochore defects in F2 embryos do not stem directly from defects in recruiting specific components required for chromosome segregation, but rather from the fact that centromere identity has not been established in the maternal germ line of the previous generation. The exact nature of correct centromere identity —which genomic regions need to be occupied—could not be determined in the context of this study, and alternative hypotheses such as a specific 3D organization of chromatin could also be envisioned. The exact stage when centromere identity is set also remains to be determined. The reappearance of CENP-A during diplotene stage (after the removal at the mitosis-to-meiosis transition; Fig 1) and the observation of chromosome segregation defects during the first mitosis in F2 embryos (Fig 5) suggests that it is established within this developmental window. The 4-h delay in rescue of the embryonic lethality in F2 embryos upon heat shock–induced expression of CENP-A FL (Figs 4D and S3) strongly argues for a role of the N-terminal tail in the germ line, and we estimated that the critical period for the presence of CENP-A FL is indeed in diplotene stage, based on the number of embryos produced in the 4 h after the heat shock. We note that CENP-A localization appears to be dynamic during the meiotic divisions [27], suggesting that centromeres could be remodeled during the maternal-to-zygotic transition. The maternal contribution of CENP-A into embryos makes it formally possible that the N-terminal tail also plays an essential role in pre-gastrulation embryos. However, it is difficult to envision why the presence of CENP-A N-terminal tail would be essential for early but not late embryonic chromosome segregation.

The CENP-A Δ-tail strain shows a *mel* phenotype. This phenotype is often explained by a maternal supply of messenger RNA (mRNA) or proteins into the embryo that is sufficient to fulfill the essential functions during development (Fig 7B). In *D. melanogaster*, many maternal effect genes affect the development of the early embryo through spatially defined depositioning of maternal RNAs or protein [68]. In *C. elegans*, many genes have been reported to show a maternal effect when mutated [69–73]. A recent study in this species also showed that maternal proteins and RNA can be inherited by a null mutant until the first larval stage [74]. We ruled out that the viability of the CENP-A Δ-tail F1 homozygotes solely depends on the presence of the maternal CENP-A FL protein (Figs 3A, 3C, and S2). Instead, it is dependent on the presence of the CENP-A FL protein in the germ line of the preceding generation (P0). We therefore hypothesize that information about centromere identity rather than CENP-A FL protein is transferred from the maternal germ line into the zygote (Fig 7B), thus representing a case of epigenetic inheritance rather than classical maternal contribution. This form of inheritance has been reported previously in *C. elegans*, often for proteins that stabilize chromatin marks [75]. One example is the polycomb repressive complex 2 (PRC2), which maintains H3K27me3 during embryogenesis [76,77]. Recently, the H3.3 chaperone HIRA has also been

implicated in setting a heritable chromatin state in the maternal germ line that can be maintained into adulthood [78]. Further work will be required to uncover how the interaction between KNL-2 and the CENP-A N-terminal tail can establish a chromatin state that defines the centromere for an entire generation.

# Materials and methods

## Nematode maintenance and strain generation

*C. elegans* strains were maintained using standard conditions at 20°C unless otherwise noted. N2 was used as a wild-type strain, and all of the genetic modifications were performed in this background unless otherwise indicated. The hT2 [bli-4(e937) let-?(q782) qIs48] (I;III) balancer allele was used to maintain the *hcp-3* (CENP-A) KO and Δ-tail alleles. Heterozygote adults of these strains give rise to 4/16 heterozygous *hcp-3* KO or Δ-tail, 1/16 homozygous *hcp-3* KO or Δ-tail, and 11/16 inviable hT2 aneuploid or hT2 homozygous offspring. S1 Table provides details for the strains used in this study. Strains generated in previous studies include VC1393 [79], GCP529 [80], CA1199, and CA1200 [36]. All insertions and deletions were generated using CRISPR/Cas-9 technology as described in [81]. RNAi experiments were performed by feeding worms with bacterial clones from the Ahringer library that produce the desired dsRNA [82].

## Immunofluorescence microscopy

For IF experiments in the germ line and embryos, young gravid adult worms were dissected in anaesthetizing buffer (50 mM sucrose, 75 mM HEPES (pH 6.5), 60 mM NaCl, 5 mM KCl, 2 mM $MgCl_2$, 10 mM EGTA (pH 7.5), 0.1% $NaN_3$). The dissected germ lines and embryos were freeze cracked and fixed in methanol for 5 min at −20°C. Slides were washed with PBS and incubated with anti-HA antibody (mouse, clone 42F13, 1:60), anti-OLLAS antibody (rat, Novus Biologicals NBP1-06713, 1:200), or anti-α-Tubulin (mouse, Sigma-Aldrich T9026, 1:1,000) overnight at 4°C. Slides were washed in PBS + 0.05% Tween and incubated with Cy3-, Alexa 488-, or Alexa 594-conjugated secondary antibodies (Jackson Immunoresearch, 1:700; Invitrogen, 1:1000) for 1.5 h at room temperature (RT). Samples were washed with PBS + 0.05% Tween, stained with DAPI, and mounted with VECTASHIELD Antifade Mounting Medium. Images were obtained using a Leica DM5000 B microscope (widefield) using 63× or 100× oil objectives (NA = 1.25 and NA = 1.30, respectively), or using a Leica SP8 microscope (confocal) using 63× and 100× oil objectives (NA = 1.40). Z-planes in confocal imaging were taken every 0.3 μm.

## Live imaging

Young gravid adults were dissected in egg buffer (118 mM NaCl, 48 mM KCl, 2 mM $CaCl_2$, 2 mM $MgCl_2$, and 25 mM HEPES (pH 7.5)), and embryos were mounted on a freshly prepared 2% agarose pad. All images were acquired using the Intelligent Imaging Innovations Marianas SDC spinning disk system, mounted on an inverted microscope (Leica DMI 6000B). Embryos were kept at 20°C and imaged with a 63× glycerin objective (NA = 1.4) and an emCCD Evolve 512 camera (Photometrics, USA). Images were captured in stacks of 10 sections along the *z* axis at 1-μm intervals every 10 s with 100 to 200 ms exposures (depending on the strain) at the 488-nm (100%) and 594-nm (100%) channels. NEBD was visually determined and used as point zero in time series analysis.

## Image and statistical analysis

All images were processed using Fiji software [83]. Z-stack and single plane images were adjusted for contrast and brightness. Additionally, for maximum z-projections of confocal images of fixed samples, a Gaussian blur filter (0.5 pixel-radius) was applied. To assess the intensity of IF signals or western blot bands, corrected total fluorescence (CTF) was calculated using the following formula: CTF = Integrated Density − (Area × Mean fluorescence of background). The significance of observed differences was tested for by using a Student $t$ test.

## Co-immunoprecipitation and western blotting

Synchronized young gravid adult worms were washed 3 times with M9, and embryos were obtained by hypochlorite treatment. Embryos were resuspended in lysis buffer (50 mM Tris-HCl (pH 7.4), 500 mM NaCl, 0.25% deoxycholate, 10% glycerol, 1% NP-40, 2 mM DTT, 1× EDTA-free protease inhibitor cocktail (Roche, Switzerland), 1xPhosSTOP (Roche)) and frozen in liquid nitrogen. Samples were sonicated with a Bioruptor Pico (Diagenode, Belgium —15 cycles, 30-s sonication, 30-s rest, snap freezing every 5 cycles) and spun down (30 min, max speed) to pellet debris. The supernatant was collected, and prewashed Pierce Anti-HA Magnetic Beads (Thermo Fisher Scientific, USA) were added. Following 2 h or overnight incubation on a rotator at 4˚C, the beads were collected using a magnetic stand and washed according to manufacturer's instructions. HA-tagged proteins were eluted by boiling the beads for 5 to 10 min in sample buffer. Protein concentrations in input samples were measured with the Bio-Rad Protein Assay (Bio-Rad, 5000006), and equal amounts were loaded on SDS-page gels. Western blotting was performed according to standard procedures using 4% to 20% gradient or 12% gels and the LI-COR Odyssey system or ECL for detection. Primary antibodies (anti-HA, Sigma-Aldrich mAb 3F10, 1:1,000; anti-OLLAS, Novus Biologicals NBP1-06713B, 1:1,000) were incubated overnight at 4˚C, and IRDye or HRP-conjugated secondary antibodies appropriate for each primary antibody were incubated for 45 to 60 min at RT.

## Yeast two-hybrid experiments

The Y2H assay was performed using the GAL4-based system as previously described by [84]. In short, the yeast strain MAV203 was transformed according to the manufacturer's instructions with pDEST22 (Invitrogen, USA) plasmids expressing transcriptional activation domain fusions or pDEST32 (Invitrogen) plasmids expressing DNA binding domain fusions. The KNL-2 cDNA fragments were cloned into pDEST22 and CENP-A cDNA fragments into pDEST32. The clones were spotted on nonselective medium (leucine and tryptophan double dropout) and on selective medium (leucine, tryptophan and histidine triple dropout with 5 mM 3-amino-1,2,4-triazole (3AT)).

## Auxin-inducible degradation (AID)

Strains for auxin-inducible degradation were constructed by introducing a degron tag at the gene locus of interest. These strains were crossed to the germline- or soma-specific TIR1 drivers described in [36], and auxin treatment was performed by transferring worms (germline depletion) or embryos (somatic depletion) to bacteria-seeded plates containing 4 mM natural auxin (indole-3-acetic acid; Alfa Aesar, Germany). The soma-specific TIR1 driver is expressed in most or all somatic tissues throughout development, from embryos to adults. Worms were kept on these plates for 4 h (germline depletion) or 3 d (somatic depletion) at 20˚C.

## Induction of transgenic CENP-A expression by heat shock

HA-tagged CENP-A was cloned under the control of a *hsp-16.2* promoter and incorporated into the worm genome using the MosSCI system [85]. In addition to the HA tag, this construct also contains an N-terminal MNase tag for chromatin endogenous cleavage (ChEC) experiments that are not described in this manuscript. Expression of this construct in adult worms was induced at 30°C for 30 min. Embryos were dissected out of the adults, either immediately after the heat shock or 1 to 8 h after recovery at 20°C. Embryos were either placed on NGM plates for the assessment of viability or fixed and stained as described above.

To estimate where in the germ line the precursors of the viable embryos observed 4 h after the heat shock were at the time of the heat shock, we determined the number of embryos produced per gonadal arm during the 4 h following the heat shock. This number can be calculated from the number of embryos present in the uterus at the time of the heat shock (x), the number of embryos laid during the 4 h following the heat shock (y), and the number of embryos present in the uterus 4 h after the heat shock (z). The average number of embryos produced after the heat shock per gonadal arm is $(y + z - x) / 2$. We found that $x = 5.8 \pm 2.1$, $y = 31.6 \pm 6.7$, $z = 5.0 \pm 1.8$. $N = 20$ adult worms in 2 independent experiments.

## In vitro purification of recombinant proteins and interaction assay

The nucleotide sequence corresponding to KNL-2 aa 269–470 (central region) and CENP-A aa 1–179 (N-terminal tail) were cloned into a vector derived from pET for expression in *E. coli* as fusion proteins containing for KNL-2 a N-terminal His$_9$-MBP tag with a tobacco etch virus (TEV) cleavage site and for CENP-A an N-terminal His$_9$-MBP tag with a TEV cleavage site and a C-terminal GST tag. The recombinant proteins were overexpressed in *E. coli* BL21-Star cells, grown in TB media at 37°C for 6 h followed by overnight induction at 18°C with 0.1 mM isopropyl-β-D-thiogalactopyranoside (IPTG). His$_9$-MBP-TEV-KNL2$^{269-470}$ induced cells were harvested by centrifugation and resuspended in TBS buffer for MBP purification (50 mM Tris-Cl (pH 8.0), 750 mM NaCl, 1 mM DTT, 0.15% CHAPS, 1 μg/ml DNase, 1 μg/ml Lysozyme), supplemented with protease inhibitors (1 mM PMSF, 1 μg/ml leupeptin, and 2 μg/ml pepstatin). His$_9$-MBP-TEV-CENP-A$^{1-179}$-GST induced cells were harvested by centrifugation and resuspended in lysis/washing buffer for NiNTA purification (50 mM phosphate buffer (pH 8.0), 300 mM NaCl, 25 mM imidazole, 0.15% CHAPS, 5 mM β-mercaptoethanol, 1 μg/ml DNase, 1 μg/ml Lysozyme), supplemented with protease inhibitors. The control GST peptide was purified from the pET42 empty vector using the same conditions as described for CENP-A$^{1-179}$. Cells were lysed using an emulsiflex system (AVESTIN, Germany). The soluble fraction was subjected to an affinity purification using an MBP FF crude column (GE Healthcare, Switzerland) for the KNL-2 fusion construct, or a chelating HiTrap FF crude column (GE Healthcare) charged with Ni$^{2+}$ ions for the CENP-A fusion constructs and the GST control. The proteins were eluted and desalted on a desalting column (GE Healthcare). For the in vitro interaction assay, the KNL-2 and CENP-A recombinant proteins were mixed in equimolar amounts. GST was used as a negative control. After 1 h incubation at RT, TEV protease was added and digestion was performed for 30 min at RT. Glutathione High Capacity Magnetic Agarose Beads (Sigma G0924) were added, and the mixture was rotated 1 h at RT. The beads were then washed and eluted, after which all fractions were loaded onto a 4% to 20% gradient SDS-PAGE gel and stained with Coomassie Brilliant Blue after electrophoresis.

## Supporting information

**S1 Fig. The CENP-A N-terminal tail interacts with the KNL-2 central domain in vitro and in vivo.** (A) In vitro interaction of CENP-A N-terminal tail and KNL-2 central domain.

CENP-A N-terminal tail (aa 1–179) fused to GST, GST alone, and the KNL-2 central region (aa 269–470) were purified from bacteria. To aid solubility, a HIS-MBP tag was fused to the KNL-2 and CENP-A peptides. The MBP tag was removed by TEV cleavage (scissors in cartoons) before the pull-down using GST beads. The GST::CENP-A N-terminal tail (left) co-precipitates the KNL-2 central fragment, while GST alone (right) does not. (B) Uncropped western blots used to generate Fig 2B. For the first repeat, samples were loaded in duplicate for detection with anti-OLLAS and anti-HA antibodies. For the second and third repeats, western blot membranes were cut at a cutoff of about 80 kDa (dashed lines), and the parts were incubated with anti-OLLAS and anti-HA antibodies, respectively. CENP-A, centromere protein A; FL, full-length; IP, immunoprecipitation; TEV, tobacco etch virus.
(TIF)

**S2 Fig. Persistence of maternally contributed CENP-A and zygotic expression of CENP-A in early embryos.** Cartoons show the different CENP-A alleles (GFP-tagged FL, green; FL, gray; KO, dark blue; Δ-tail, orange) in heterozygotes (striped) or homozygotes (full). Dashed boxes highlight the embryos relevant for the analysis. In the fluorescence images, arrowheads point to embryos with visible CENP-A FL protein, arrows point to embryos without visible CENP-A FL protein, and asterisks mark embryos with zygotically expressed CENP-A FL protein that obscures the analysis of maternal contribution. (A) Crossing males homozygous for GFP- and HA-tagged CENP-A FL to feminized *fem-2* hermaphrodites homozygous for untagged CENP-A FL results in heterozygous F1 progeny that allow detection of the onset of zygotic expression of CENP-A (top). Self-fertilization of these F1 heterozygotes results in 25% embryos homozygous for untagged CENP-A FL that allow determination of the limits of maternal CENP-A FL contribution (bottom). (B) Determination of maternal CENP-A FL contribution as in (A), but CENP-A FL is OLLAS-tagged instead of GFP- and HA- tagged and is analyzed in the context of CENP-A KO or CENP-A Δ-tail instead of untagged CENP-A FL. Self-fertilization of heterozygotes results in 25% embryos homozygous for untagged CENP-A FL that allow determination of the limits of maternal CENP-A FL contribution. Scale bars represent 20 μm in (A) and (B). (C) Summary of the analysis described in (A, B), showing that maternally contributed CENP-A protein is detectable until the 100–200 cell stage, and zygotic expression is first detected at the 30–40 cell stage. Embryonic stages from beginning (26 cells) to end (330 cells) of gastrulation reflect those shown in the Wormatlas (www.wormatlas.org). CENP-A, centromere protein A; FL, full-length; KO, knockout.
(TIF)

**S3 Fig. Detection of heat shock−induced CENP-A FL protein by IF.** Pachytene and diakinesis nuclei of adults and embryos (highlighted in the cartoon image of the gonad) before and at the indicated time points after heat shock are shown. The images are shown for heat shock−induced HA-tagged CENP-A FL protein in a non-tagged CENP-A FL/FL background, because the CENP-A Δ-tail protein in the strain used for Fig 4D is also HA-tagged. Scale bars correspond to 5 μm. CENP-A, centromere protein A; FL, full-length; IF, immunofluorescence.
(TIF)

**S4 Fig. Mitotic and meiotic segregations are differentially affected in CENP-A Δ-tail homozygous F2 embryos.** (A) Altered KNL-2 dynamics in mitosis of CENP-A Δ-tail homozygous F2 embryos. IF images depicting KNL-2 and α-tubulin at metaphase of early embryonic cell divisions in embryos derived from CENP-A FL/FL, FL/Δ-tail, or Δ-tail/Δ-tail maternal germ lines. (B) Meiosis is unaffected in CEN-P-A Δ-tail homozygous F2 embryos. Top, cartoon images of embryos with the chromosome stages corresponding to the live cell images in red. Bottom, still images of live cell recordings of embryos derived from CENP-A FL/Δ-tail or

Δ-tail/Δ-tail maternal germ lines. H2B::mCherry was used to visualize chromosome segregation in meiosis I and meiosis II and mitosis of the first embryonic cell division. Polar bodies are labeled with PB and arrows. CENP-A FL/Δ-tail and Δ-tail/Δ-tail F1 offspring are indistinguishable as early embryos; therefore, the F1 genotype is marked with an asterisk. Scale bars represent 2 μm. CENP-A, centromere protein A; FL, full-length; IF, immunofluorescence; WT, wild-type.
(TIF)

**S5 Fig. CENP-A and KNL-2 patterns in the germ line of worms with different CENP-A genotypes.** IF images showing the patterns of KNL-2 and CENP-A FL and Δ-tail, counterstained with DAPI, in adult germ lines of CENP-A FL/FL (WT control), FL/Δ-tail (P0), and Δ-tail/Δ-tail (F1) animals. (A) Whole gonads dissected from adults, and cartoon describing the different regions of the gonads. The extension of distal CENP-A Δ-tail and KNL-2 signal into the pachytene region in CENP-A Δ-tail homozygotes is visible in 43% of the gonads analyzed. Yellow boxes highlight the regions shown in (B). The dashed lines outline the gonads, and the dotted lines indicate where 2 images of the same gonad have been merged. (B) Zoomed images of nuclei in the mitotic zone, the pachytene region, and diakinesis oocytes, as highlighted in the cartoon image of the germ line in (A). Yellow boxes highlight individual nuclei that are enlarged below. Arrows indicate the KNL-2 foci in diakinesis. Scale bars represent 20 μm in (A) and 5 μm in (B). CENP-A, centromere protein A; FL, full-length; IF, immunofluorescence; WT, wild-type.
(TIF)

**S6 Fig. Comparison of CENP-A FL and Δ-tail levels.** CENP-A levels were compared in CENP-A FL or Δ-tail homozygous (A, B) and in CENP-A FL/Δ-tail heterozygous (C, D) animals. In homozygous animals, CENP-A FL or CENP-A Δ-tail is HA-tagged, whereas in heterozygous animals, one copy of CENP-A FL is OLLAS-tagged, and the other copy is either HA-tagged CENP-A FL or HA-tagged CENP-A Δ-tail. CENP-A levels were determined and quantified in diakinesis nuclei (A, C) and on metaphase plates in early embryos (B, D). $N = $ 27–43 oocytes and 13–18 embryonic metaphase plates from 3 independent IF experiments. In each IF experiment, the mean level of CENP-A FL in CENP-A FL homozygotes was set to 1. Error bars show the standard deviation of the mean. Asterisks denote statistical significance, determined by using a Student $t$ test. Scale bars represent 5 μm. The data underlying all the graphs can be found in S1 Data. CENP-A, centromere protein A; FL, full-length; IF, immunofluorescence.
(TIF)

**S7 Fig. Embryonic lethality upon depletion of LIN-53.** Cartoons show the different CENP-A alleles (FL, gray; Δ-tail, orange) in heterozygotes (striped) or homozygotes (full). Percentage of expected hatchlings in control animals and upon LIN-53 depletion in CENP-A FL homozygous and CENP-A FL/Δ-tail heterozygous mothers are shown. RNAi-mediated depletion of LIN-53 was induced by feeding in parental animals from L4 larval instar. CENP-A FL/Δ-tail heterozygotes contain the balancer allele, and CENP-A FL homozygous embryos are therefore inviable. $N = $ 17–20 broods from 3 independent RNAi experiments. Asterisks denote statistical significance, determined by using a Student $t$ test. The data underlying all the graphs can be found in S1 Data. CENP-A, centromere protein A; FL, full-length; RNAi, RNA interference.
(TIF)

**S1 Table. *C. elegans* strains generated or used in this study.**
(XLSX)

**S1 Video. KNL-2::GFP in first embryonic division for embryos derived from CENP-A FL/Δ-tail maternal germ line (F1 embryo).**
(MP4)

**S2 Video. KNL-2::GFP in first embryonic division for embryos derived from CENP-A Δ-tail/Δ-tail maternal germ line (F2 embryo).**
(MP4)

**S3 Video. mCherry::H2B and mCherry::γ-tubulin in first embryonic division for embryos derived from CENP-A FL/Δ-tail maternal germ line (F1 embryo).**
(MP4)

**S4 Video. mCherry::H2B and mCherry::γ-tubulin in first embryonic division for embryos derived from CENP-A Δ-tail/Δ-tail maternal germ line (F2 embryo).**
(MP4)

**S5 Video. ROD-1::GFP in first embryonic division for embryos derived from CENP-A FL/Δ-tail maternal germ line (F1 embryo).**
(MP4)

**S6 Video. ROD-1::GFP in first embryonic division for embryos derived from CENP-A Δ-tail/Δ-tail maternal germ line (F2 embryo).**
(MP4)

**S7 Video. HCP-4::mCherry in first embryonic division for embryos derived from CENP-A FL/Δ-tail maternal germ line (F1 embryo).**
(MP4)

**S8 Video. HCP-4::mCherry in first embryonic division for embryos derived from CENP-A Δ-tail/Δ-tail maternal germ line (F2 embryo).**
(MP4)

**S1 Data. Numerical values for figures as indicated in the individual sheets.**
(XLSX)

## Acknowledgments

We thank members of the Steiner lab for help and discussion, Dario Menéndez for help with strain generation, Marina Berti for reagent preparation, Nicolas Roggli for help with the figure preparation, Patrick Meraldi and Nikolai Klena for comments on the manuscript, and Stella Toonen for proofreading. Some strains were provided by the CGC, which is funded by NIH Office of Research Infrastructure Programs (P40 OD010440). We are grateful to the Bioimaging Center of the Faculty of Sciences at the University of Geneva.

## Author Contributions

**Conceptualization:** Reinier F. Prosée, Florian A. Steiner.

**Formal analysis:** Reinier F. Prosée, Florian A. Steiner.

**Funding acquisition:** Monica Gotta, Florian A. Steiner.

**Investigation:** Reinier F. Prosée, Joanna M. Wenda, Isa Özdemir, Caroline Gabus, Francoise Schwager.

**Methodology:** Reinier F. Prosée, Joanna M. Wenda, Isa Özdemir, Caroline Gabus, Kamila Delaney, Francoise Schwager, Florian A. Steiner.

**Project administration:** Reinier F. Prosée, Florian A. Steiner.

**Resources:** Monica Gotta, Florian A. Steiner.

**Supervision:** Monica Gotta, Florian A. Steiner.

**Validation:** Reinier F. Prosée, Joanna M. Wenda, Isa Özdemir, Caroline Gabus, Francoise Schwager.

**Visualization:** Reinier F. Prosée, Florian A. Steiner.

**Writing – original draft:** Reinier F. Prosée, Florian A. Steiner.

**Writing – review & editing:** Reinier F. Prosée, Joanna M. Wenda, Isa Özdemir, Kamila Delaney, Monica Gotta, Florian A. Steiner.

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
