## [Editor Report · Decision Letter 0]

29 Sep 2020

Dear Dr Steiner, 

Thank you for submitting your manuscript entitled "Trans-generational inheritance of centromere identity requires the CENP-A N-terminal tail in the C. elegans maternal germ line" for consideration as a Research Article by PLOS Biology.

Your manuscript has now been evaluated by the PLOS Biology editorial staff as well as by an academic editor with relevant expertise and I am writing to let you know that we would like to send your submission out for external peer review.

Please re-submit your manuscript within two working days, i.e. by Oct 01 2020 11:59PM.

Kind regards,

Ines

--

Ines Alvarez-Garcia, PhD

Senior Editor

PLOS Biology

---

## [Decision Letter · Decision Letter 1]

26 Nov 2020

Dear Dr Steiner,

Thank you very much for submitting your manuscript entitled "Trans-generational inheritance of centromere identity requires the CENP-A N-terminal tail in the C. elegans maternal germ line" for consideration as a Research Article at PLOS Biology. Thank you also for your patience as we completed our editorial process, and please accept my apologies for the delay in providing you with our decision. Your manuscript has been evaluated by the PLOS Biology editors, an Academic Editor with relevant expertise, and by three independent reviewers.

You will see that all three reviewers agree that the finding that CENP-A has different roles in the germ line and embryo is interesting. Nevertheless, both Reviewers 2 and 3 raise several concerns regarding the clarity of the text and the fact that some of the conclusions are not fully supported by the data. They suggest experiments aimed to address these issues that you should perform.

In light of the reviews (attached below), we will not be able to accept the current version of the manuscript, but we would welcome re-submission of a revised version that takes into account the reviewers' comments. We cannot make any decision about publication until we have seen the revised manuscript and your response to the reviewers' comments. Your revised manuscript is also likely to be sent for further evaluation by the reviewers.

We expect to receive your revised manuscript within 3 months. 

**IMPORTANT - SUBMITTING YOUR REVISION**

*Re-submission Checklist*

*Published Peer Review*

*PLOS Data Policy*

*Blot and Gel Data Policy*

Sincerely,

Ines

--

Ines Alvarez-Garcia, PhD

Senior Editor,

PLOS Biology

Reviewers’ comments

Rev. 1:

This paper investigates the very important issue of how centromeres are assembled and maintained in the germline and how they are inherited from one generation to the next. Specifically, this paper investigates the CENP-A inheritance in worms. Worms display unusual properties with respect to CENP-A inheritance. Firstly, sperm (paternal) chromosomes are devoid of CENP-A and it is reloaded only after fertilisation. Secondly, CENP-A is removed at the mitosis-to-meiosis transition the germline and it is reloaded in diplotene of meiosis. Therefore, it is apparent that different from 'template' models of inheritance that have been proposed in other organisms, centromeres must be established de novo at both of these stages in worms. Here the authors convincingly show a key role for the CENP-A N terminus, as well as KNL2 and LIN-53, in de novo centromere assembly in the germ line.

I have very little criticism of this paper. It is a very interesting read and presents some unexpected results in a clear and concise manner. All experiments are well designed, well executed and results are well interpreted. The results have far reaching implications with respect to mechanisms of epigenetic inheritance in the germline.

I have only three very minor queries/comments:

- In Figure 2, the authors convincing show that the CENP-A N terminus is both necessary and sufficient for interaction with KNL2 using both in vitro and in vivo approaches. In Figure 2B, the error bar for KNL2 enrichment in the CENP-Aaa1-99 pulldown is quite large and only two replicates were performed. The authors should clarify how this enrichment was calculated. In Figure 2C, lanes 4 and 5 appear to be switched. Shouldn't FL-CENP-A run higher than the truncated version?

- In Figure 4B, the authors perform live imaging of the first embryonic division in WT, F1 and F2 embryos. Can the authors comment on the observed defects in chromosome condensation which are visible in the F2 embryos and perhaps also in the F1 embryos? Is this phenotype expected to result from a lack of centromere specification?

- In Figure S4, the authors investigate CENP-A and KNL2 localisation in the delta tail/tail F1 germline. Both proteins display a punctate pattern in the mitotic zone and in diakinesis. Can the authors comment on whether it is the intensity of these foci that has increased or whether a relocalisation event has occur? Increased intensity might indicate enhanced assembly or stability of CENP-A/KNL2 at this time, which could be important for understanding the mechanism of de novo centromere establishment.

Rev. 2:

Prosée et al report two main findings: 1. That a deletion of the N-terminal tail of the centromeric histone CENP-A is a maternal effect lethal allele and 2. That the defect that causes this embryonic lethality is a failure in specifically KNL-2 mediated de novo centromere establishment in the maternal germline. The role of the N-terminal tail in germline transmission has been reported in plants, where an N-terminal CENP-A tailswap with the conventional H3 tail results in uniparental elimination of the genome with the mutant CENP-A (Ravi and Chan, 2010). The difference is that in this case, the tail mutant of CENP-A appears to have a phenotype in the following F2 generation, typical of a maternal effect allele. This is an interesting observation, but the interpretation of the data remains unclear in some areas. The crucial data that demonstrates the requirement of re-establishment of centromeres in the maternal germline for F2 viability is in the supplemental section which gives me pause since that is critical for the main conclusions (indeed, needed to support the claims of the title of the paper!). In addition, the authors find that LIN53 (previously described as a general histone chaperone that also assists in CENP-A nucleosome assembly, Ref 31) is also involved in CENP-A chromatin association but this is not well developed in the paper. As it stands the paper has an interesting observation but to highlight its importance, the paper would require substantial restructuring, more compelling data interpretation, and additional clarification of aspects of the experimentation (one example are some of the key images in Figure 4B; see point #5, below). Specific comments to elaborate on this overall assessment are as follows:

Major comments:

1. The major claim of the paper is that the N-terminal tail of CENP-A is dispensable for mitosis during development but required for re-establishment in the maternal germline. However, in the absence of an initial template supplied by wild type maternal CENP-A (FL) templated chromatin, the N-terminus is also unable to support mitosis (since it apparently doesn't know where to assemble), so the term dispensable is confusing here (Fig 3A, C). Further, the somatic auxin depletion (3-day treatment in adults, S2B) does not affect hatch rates and could also be explained as the already established epigenetic identity in the P0 germline being maintained. It would seem quite possible that the auxin depletion is not complete and the remaining CENP-A, however little, is sufficient to direct the mutant to the right genomic locations. A western to quantify the auxin depletion in the germline vs somatic depletion would help clarify this. Figure 3A, C and S2 are the most impactful observations in the paper in my opinion and should be further highlighted in the text. It would also be stronger, in my opinion to present them together in one main figure. The distinction between the germline re-establishment vs. maintenance made by the somatic vs germline depletion experiments and the F1 sterile hermaphrodites should ideally be more clear in a revised manuscript.

2. All the experiments point towards a defect in de novo establishment of maternal centromere identity in the �Tail mutant F1 germline. This is the crux of the paper (in the title) and the new advance that makes this paper different than the entire bulk of the plant studies on the N-tail. Therefore, it is quite surprising that the data that specifically demonstrates this is a supplemental figure (S5)! I strongly suggest moving that data into the main paper. Just looking at Fig S5 I do not see a clear difference between the P0 and F1 germline CENP-A other than the punctate nature of KNL-2. Since this is unclear, it would seem to be crucial that the authors provide images of the full germline (as in Figure 1) for all the genotypes in S5, specific nuclei in mitotic and diakinesis zones can be zoomed in on in insets. This will also help for the reader to appreciate the claim in line 290-292 that the CENP-A mutant removal is delayed. The authors have convincing data that this chromatin associated CENP-A is clearly not functioning in centromere identity since the outcome is drastic mis-segregation later in the embryo. But it is very confusing as to why the next section in the text after referring to S5 assumes that KNL-2 is required for de novo assembly in the germline when the diakinetic nuclei clearly have chromatin associated CENP-A. The authors explain that the KNL-2 mediated CENP-A assembly is the functional population which is clear from the lethality in the F2 generation but I am not sure why LIN53 mediated chromatin association is not regarded as assembly but is nonetheless designated as "maintenance". Since CENP-A is removed early in the germline, any chromatin association in the proximal germline has to be de novo assembly and not maintenance (see point 3, below).

3. The title of Figure 5 "Chromatin association of CENP-A depends on LIN53 and not KNL-2' is confusing. What do the authors mean by chromatin association? All nucleosomes are chromatin bound including the KNL-2 population apparent from the fact that the P0 and F1 germlines do not look different in CENP-A localization unless the levels are lower. I think what the authors mean is that the KNL-2 assembled population is the only functional population at least for de novo establishment of centromere identity. The result in this section brings up the interesting idea that even in holocentrics the location of CENP-A matters. The KNL-2 mediated de novo assembly (Figure 5) must be directed to certain locations perhaps more conducive for establishing epigenetic memory whereas the LIN53 mediated assembly could be more random; this should be discussed in the text. Does germline specific knockdown of lin53 yield viable embryos? Do the F1 embyros obtained by treating FL/�Tail P0 hermaphrodites with lin53 RNAi develop?

4. Quantifying the levels of CENP-A, like with the embryos in Fig 5 A and B, to better understand the two pathways in the germline where the action happens would help support the conclusions made in the paper. Quantification of the levels of FL vs �tail in the P0, F1 and F2 embryos is also going to be informative. Is the �Tail just generally less efficient in nucleosome assembly?

5. In figure 4B, unless I am confused by what is shown, there appears to be an image repetition in the metaphase panel for the homozygous F2 �tail mutant embryos. The GFP::KNL-2 and the GFP::ROD1 panels look nearly identical for those mutant embryos.

6. Shouldn't the model include LIN53 mediated maintenance since it appears to be important for the downstream function of the centromeres once KNL-2 mediated de novo assembly is established in the germline? In addition, although Figure 6B is explained well in the text, the figure does not demonstrate the germline epigenetic inheritance posited by the authors. The authors provide two scenarios, function vs presence of protein but a conclusion consistent with their experiments (and those of many others) is that epigenetic inheritance is essentially the presence of protein, just assembled correctly in the genome versus existing as a maternal pool. I think the authors would like to make the distinction between maternal contribution (classical maternal effect genes) versus maternal epigenetic inheritance, which refers to all chromatin dependent epigenetic marks inherited through the parent. This should be clarified, as the defect the authors want to highlight is in the maternal germline epigenetics and not a consequence of defective maternal protein pool in the embryo.

7. The inclusion of the recombinant protein data (Fig. S1) is questionable. If it was robust and well controlled it would be in the main figure set. As is, it seems unreliable and so it is relegated to the supplement. This doesn't seem like a strategy that engenders a lot of faith in the conclusions of the paper more broadly.

Minor comments:

8. N should be listed throughout. It is unclear what the dots represent in many places. The data are all provided as percentages, so the actual number of worms must be added to all relevant figures including the number of repeats (or P0 parents used). In addition, for the somatic depletion of FL protein, when was the counting performed, at the end of 3 days?

9. It would also help broad non-germline readers if the schematic for germline development is put in the context of when maternal contribution ends in the worm. This would make it clear that there is no maternal FL protein available in the F1 homozygote germline. This point is mentioned in Line 195-197 but citing Figure 3B-F1 diakinetic nucleus would also help.

10. Suggested re-organisation of the paper: Perhaps the authors could consider moving S5 to Figure 4 and Figure 4 to S5. Figure 4 currently enumerates how the embryos are dying by mis-segregation, which is important but expected for a CENP-A mutant. The main result is that the defect is upstream in the female germline and should be highlighted in the paper.

11. All nuclear fluorescence quantifications should be normalized to the control (WT and/or without auxin) so as to make it easier for the reader to gauge levels of decrease.

12. For table S1, it would be really helpful if the strains used in the different figures were annotated in the "Comments" column. Also, the germline and soma specific drivers used should be specified there for easy access for the reader.

13. In the discussion of the paper, it would be nice to discuss these results in the context of and contrasted to the plant N-terminal tail studies and highlight how this instance compares (elaborate lines 398-401). Also, the N-terminal tail of mammalian CENP-A has also been shown to be dispensable (if other parts of CENP-A are present) for maintenance (Ref 51) but indispensable for establishment at an ectopic site (Ref 52). It would seem to help the author's case about broad biological implications of their findings to make more mention of all of these prior findings.

14. Line 277 sentence needs references and to remind again which systems CENP-A has been shown to be dispensable for meiosis.

15. Line 346: I do not believe Mis18 was tested in Ref 25.

Rev. 3:

The paper set out to study how CENP-A is inherited trans-generationally. This paper presents work carried out in the nematode C. elegans to identify the N-terminal tail of CENP-A as a critical domain for the interaction with the conserved kinetochore protein KNL-2, and argued that this interaction plays an important role in setting the centromere identity in the germ line (diakinesis stage).

They first used yeast two-hybrid assay to investigate the interplay between CENP-A and KNL-2 and found that aa 1-122 of the CENP-A N-terminal tail are sufficient for the interaction with KNL-2. In vivo, CENP-A N-terminal tail (aa 1-99)-HIS-72 (a non-essential histone H3.3 homologue) chimeras was sufficient to mediate interaction with KNL-2 in co-IP experiments.

Then they constructed and maintained a CENP-A N-terminal tail truncation allele heterozygous worm strain in the context of a full-length copy of CENP-A balancer allele. They used this balancer strain (CENP-A FL/∆-tail) P0, the F1 and F2 embryos and germ line to investigate the function of the N-terminal tail, the interaction with KNL-2, and the localization interdependency with KNL-2.

They found that the F1 generation of the balancer strain homozygous for CENP-A ∆-tail were fully viable and developed indistinguishably from wild type worms and showed no obvious germline proliferation defects, while the F2 embryos laid by the homozygous F1 were all inviable due to severe chromosome segregation defects. So, they conclude that the N-terminal tail of CENP-A is dispensable during development (embryogenesis), but is required in germ line for de novo centromere establishment in each generation. However, the presence of FL CENP-A in ∆-tail homozygous F1 embryos (Fig 3B) makes it difficult to make the conclusion (see major comments).

By using the AID system, they selectively depleted germline full-length CENP-A in P0 and observed severe chromosome missegregation in all F1 embryos. Depleting the P0 somatic CENP-A deos not cause defect in ∆-tail homozygous F1. Crossing in a CENP-A full-length gene copy through the paternal germ line does not rescue F2 embryonic lethality. With these results they conclude that maternally provided full-length CENP-A is required for de novo centromere establishment in the maternal germ line.

In the Diakinesis oocyte (proximal germ line), FL CENP-A loading is dependent on KNL-2, but not vice versa, suggesting that KNL-2 acts upstream of CENP-A in the de novo establishment of centromeres. Interestingly, CENP-A ∆-tail remains on chromatin upon depletion of KNL-2 in F1 CENP-A ∆-tail homozygotes in a KNL-2-independent manner. However, it is still unclear how the N-terminal CENP-A and KNL-2 contribute in de novo centromere establishment in detail. In addition to immunofluorescence, if the authors can show how the CENP-A ∆-tail becomes mislocalized on the genome (or mislocalized with other centromeric proteins), it may give more clue to the mechanism.

Through the elegant genetic design of the balancer strain (CENP-A FL/∆-tail), AID strains and accompanied biochemistry and imaging analyses, this study has helped to elucidate the interaction of KNL-2 with full-length and truncated CENP-A in vivo, and also the function of this interaction in CENP-A localization in the C. elegans proximal germline and the subsequent mitosis. In summary, it is exciting to see this work trying to elucidate the different role of CENP-A, KNL-2 and their interaction in the germ line vs. in embryos. However, it is not entirely convincing. If the authors can tone down on their conclusion on the germline specificity and on the "functional inheritance" mechanism, and just focused on their solid interaction and dependency results, it is still an interesting study with great contribution to the field. The observed phenotypes provoke insights and deserve further mechanistic investigation. Further work will be required to elucidate why and how the interaction between KNL-2 and the CENP-A N-terminal tail can establish de novo centromere in germline and how to transmit this centromere identity into the next generation. Some questions are listed below.

Major:

For the co-IP experiment, the authors used embryo extract. However, if the authors are mainly interested in de novo centromere establishment in the diplotene stage, shouldn't they also isolate germline extract (or use sterile fem-1 mutants?) to perform the co-IP?

Page 9 line 205 to 213

I do not understand how the AID experiment can exclude the possibility that "undetectable amounts of full-length CENP-A that could support development" in the �N-tail homozygous F1 that is inherited from P0. The AID experiment shows that if FL is depleted in P0 germline, Degron::FL/�tail F1 cannot hatch (no detectable FL in heterozygous diakinetics). What is the signal in Fig 3D FL CENP-A?

On the other hand, if FL is depleted only somatically, Degron::FL/�tail F1 can hatch (Fig. S2B quantification, but no IF image). For somatic depletion, are P0 in adult stage and what promoter is used? It means there will be maternal load FL CENP-A in F1 embryos? So again, it is not too surprising that they survive.

Doesn't Fig. 3B show that �N-tail homozygous F1 embryos have detectable level of FL by IF, and cannot be distinguished with het? This hinders me from believing that the difference in F1 and F2 �N-tail homozygous is only due to the germ line.

Although the authors mentioned that a strain containing a balanced CENP-A deletion produces homozygous CENP-A deletion that is embryonic lethal. This is a more fair comparison to imply the function of �N-tail in embryos. As there is no ref, are the balancers the same. Have the authors performed IF of embryos to show the maternal load to germline and F1 embryos are really the same in �N-tail homozygous and this CENP-A deletion?

Page12 line286

"Germ line of F1 CENP-A �N-tail/�N-tail animals, both CENP-A and KNL-2 show a punctate pattern, even though mitotis appears to be normal in these cells (S5 Fig)." The"punctate" pattern is apparent in the mitotic zone of germline (S5 Fig). However the CENP-A �N-tail in diakinesis stage is not obviously different. How is mitotic accuracy in mitotic tip observed (Fig 4A)? How do the authors envision the punctate pattern (instead of on most of chromatin) affect centromere establishment/identity/localization?

Indeed, is it true that without the CENP-A N-tail interaction domain with KNL-2, the localization of CENP-A N-tail remains the same in diakinesis oocyte in Fig. 3B (with KNL-2) and in Fig. 5B (in KNL-2 depletion)? And what is the expected effect on centromere establishment/identity/localization?

Page 12 line 290 to 292

"Moreover, the removal of both CENP-A and KNL-2 at the mitosis-to-meiosis transition is delayed, and both proteins can be detected in part of the pachytene region of the germ line." The result or image cannot be found.

Page 18 line 441 to 446

"We ruled out that the viability of the CENP-A ∆-tail F1 homozygotes solely depends on the presence of the maternal full-length protein. Instead, it is dependent on the presence of the full-length protein in the germ line of the preceding generation (P0). We therefore hypothesize that information about centromere identity rather than full-length CENP-A protein is transferred from the maternal germ line into the zygote (Fig 6B)." The authors hypothesize that it's centromere identity rather than the full-length CENP-A protein is inherited from germline to zygote, but the centromere identity in germline needs full-length CENP-A. It is unclear what could be the nature of the information of the identity that transmit? Based on Figure 6A, how does KNL-2 bring FL CENP-A to the correct genomic location in this establishment process?

Similarly, in Figure 6B what exactly is the meaning of "functional inheritance". In this model, F1 embryos still have residual materanla proteins also? If so, how can it be distinguished by the protein/RNA model? Can the authors at least propose what functional inheritance is mediated by? Histone modification?

Ultimately, how can the authors attribute the F2 embryo chromosome missegregation to F1 germ line reestablishment, and not just no FL CENP-A in F2 embryos? The maternal load effect influence both germ line and the embryos produced. It may require a system that remove CENP-A in either the embryos or diakinesis, respectively. Is this feasible in the AID system developed here with early embryo re-expression/complementation?

Minor:

Fig. 2B why the input KNL-2::OLLAS look so variable? While it is a 2-fold difference between 1 and 2, does the error bar suggests that they are not significantly different?

Fig. 2C blot seem to have bubble and distortion of band? No statistics?

p. 8 line 198 Please use CENP-A or hcp-3 consistently.

Fig. 4A �N-tail/�N-tail F2 embryos KNL-2 cluster as big dots, but in Fig. 4B KNL-2 is diffused in prometaphase and metaphase, but have dots in anaphase. Can you explain the discrepancy? Can the authors propose why the mislocalized KNL-2 is at/with the microtubules?

p. 11 line 273 "We attribute these F2 embryonic phenotypes to the failure in establishing centromere identity in the germ line of F1 homozygotes (CENP-A �N-tail/�N-tail)." I think it is a premature attribution or conclusion.

Page 12 line 277 "CENP-A has previously been shown to be dispensable for meiosis" needs a citation. Ref 28?

Page 13 line 304 "depleting FL CENP-A in the proximal germ line did not lead to loss of KNL-2, as was reported in embryos" needs a citation for the embryo observation.

Page 13 line 308 The authors are discussing KNL-2-independent CENP-A �N-tail localization, whereas the previous sentence is about CENP-A-independent KNL-2 localization. So both are indistinguishable from wild-type?

Page 15 line 364 "However, non-mammalian KNL-2s often show cell cycle dynamics that are different……". Should it be "non-mammalian homologues of KNL-2"?

Page 19 line 468 to 469 "RNAi experiments were performed using clones from the Ahringer library [80]". Although given the citation, the authors should at least mention that the RNAi experiments are performed by feeding RNAi. Non-worm people may be confused why the RNAi experiments were performed by using clones from a special library.

Page 20 line 484 to 485 "Images were obtained using a Leica DM5000 B microscope." Can the authors provide information on the use of objective and z-stack acquisition.

Page 20 line 489 to 490 "Young gravid adults were dissected on a 2% agarose pad and kept in egg buffer". Should it be "Young gravid adults were dissected in egg buffer and embryos were mounted on a freshly prepared 2% agarose pad"?

Have the authors looked at where the CENP-A N-terminus-H3.3 chimera localizes, and whether it complements some function of CENP-A?

---

## [Decision Letter · Decision Letter 2]

7 May 2021

Dear Dr Steiner,

Thank you for submitting your revised Research Article entitled "Trans-generational inheritance of centromere identity requires the CENP-A N-terminal tail in the C. elegans maternal germ line" for publication in PLOS Biology. I have now obtained advice from the two of the original reviewers and have discussed their comments with the Academic Editor. 

Based on the reviews (attached below), we will probably accept this manuscript for publication, provided you satisfactorily address the remaining points raised by the reviewers. Please also make sure to address the following data and other policy-related requests.

We expect to receive your revised manuscript within two weeks. 

*Published Peer Review History*

*Early Version*

Sincerely,

Ines

--

Ines Alvarez-Garcia, PhD,

Senior Editor,

PLOS Biology

DATA POLICY:

Thank you for complying with our data policy. Please also ensure that figure legends in your manuscript include information on where the underlying data can be found, and ensure your supplemental data file/s has a legend.

DATA NOT SHOWN?

Reviewers’ comments

Rev. 2:

The revised submission is much improved and the authors responded to the initial critiques in a positive way, including the addition of new data. The clarity and flow of the paper including data interpretation and presentation is better. The added experiments help better support the major claims in the paper. There are, however, a few minor points for the authors to consider in order to help clarify some of the new data:

1. Can the authors comment on why Figure 4A appears to show a lower percentage of the heterozygous male progeny compared to the homozygotes? Shouldn't it be 50-50 ratio as in 4B? Appears to be consistent for all broods which would argue against experimental variation.

2. The heat shock experiment is an elegant way of showing the stage specific assembly requirement in the maternal germline. The degradation of CENP-A obviously occurs outside of the time window (~4 hours) when de novo centromere establishment happens. Does the observed time frame of 4 hours coincide with diplotene to embryo developmental time as expected? This would be worth discussing and explaining further in the text (related to Figure 4D and S3).

3. Figure 4C should have the generation (F2) also annotated directly in the panel, since 4B also has the same genotype but no particular generation.

4. As the authors point out in lines 419-427, Lin53 plays a role in both maintenance and possibly establishment as it reduces the levels of both FL and delta tail CENP-A. But the authors could add that this reduction is more severe than that of KNL-2 depletion alone and hence provides further support for a role in 'maintenance'.

Rev. 3:

This article examines the centromere protein A N-terminus function in centromere identity inheritance across generations and distinguishes that from CENP-A's mitotic function in mediating accurate chromosome segregation. In this revised version, the author used CENP-A KO to contrast with CENP-A N-tail deletion. They think that the relatively normal F1 is maintained by CENP-A N-tail deletion expressed in germline, as the counterparts with CENP-A KO don't exhibit the same behavior in F1. Also the AID system was used to show 2 points: 1/ the CENP-A N-del is able to support mitosis and development, 2/ the CENP-A N-del is unable to reload after pachytene (at some %?), but heat shock-mediated transient CENP-A FL expression (in diakinesis but not simply in embryos) can rescue the embryonic lethality. The manuscript has improved a lot with these additional experiments. However, there are multiple instances that the responses to reviewers' questions are much more clear and straightforward (e.g. major comment 4 on the alternative hypothesis, and major comment 9 on the summary of the findings and the proposed epigenetic mechanism) than the wordings used in main text. The authors can consider using the justifications in text. Also for the minor suggestions below, while readers can probably guess what the authors mean, it will be better if the authors express more clearly what they mean explicitly.

Major:

Figure 4D. There is a 4-hour time frame between CENP-A FL expression and the rescue in early embryo segregation and development. What stage was it 4-hours before the first mitotic division? To non-worm audience, it is quite difficult to translate the 4-hour delay to developmental time frame. In addition to citing S3 Fig, could the authors explain this in the main text?

In Line 570, the author considered this delay strongly argues for a role of N-terminal tail in the germline rather than in the embryos. However, in S3 Fig, 0 h also has FL in diakinesis already. How long does it take from diakinesis to embryos? In other words, show for the "0 hr embryos", there is no FL x hrs earlier in the diakinesis. This is one of the most important added experiment and is worth describing more clearly.

Minor:

CENP-A is absent from centromere during pachytene, and indeed it is also absent in the paternal chromosomes before fertilization. They want to find out how CENP-A re-localizes to the centromere after CENP-A is removed from the chromatin in pachytene. They used the term "de novo establishment of centromeres" (Line32) to describe the centromere/CENP-A reappearance/re-establishment in meiosis I diplotene. The authors can consider using "re-" instead of "de novo".

Figure 2B. + S1. Fig

In 2nd repeat of the IP triplicates, OLLAS (KNL-2) signal is weak for both tail and FL. However, in the main figure quantification, error bars in Figure 2B are small for KNL-2 enrichment. Specify the N, and if 2nd repeat in S1 Fig is excluded in quantification in Figure 2B, maybe it should be removed from the fig. (Previously commented as Q11, author responded: technical problem, redo and excluded that set of data from qualification.)

Figure 3A

In the legend, "quantification of the viable offspring as a percentage of the expected hatchlings in the context of the balancer allele". Do the author mean that D-tail/balancer heterozygotes's hatching is set as 100%, and other genotypes are compared to this? If so, such normalization is good, but the figure Y-axis "% expected hatchings (balancer)" is very hard to understand. Maybe just say "% hatchings (normalized to balancer het)"?

Figure 3B, 3D

Instead of * and explaining it in the legend the indistinguishable genotypes, the authors may as well put the two genotypes in the figure for clarity.

Line 191, KO/KO "presumably receives the same dose of CENP-A FL....as those produced by CENP-A D-tail strain (Fig 3A)" It would be nice if this is actually shown by IF, to show that the maternal load (level) is not affected by the genetic background (or refer to S2B Fig here?). Line206-210 seems more relevant here.

Lin3 224, Figure 3C, can the authors briefly mention when is the soma-specific eft-3 promoter ON (or when FL is degraded) for the non-worm audience?

Figure 3D, line 257

FL OLLAS staining in F1 embryos is from FL/D-tail zygotic? So indeed the 2 genotypes can be distinguished at this (later) stage of embryos? Then can show them separately instead of using *. But why is the signal not in all nuclei, but only in some?

Line 325, F2 embryos show "decondensation defects (Fig 5A). First, the defects should be "condensation defect"? Also, no condensation/decondensation assay was performed in this study. They can simply describe the shape of chromatin without inferring to the condensation process.

Figure 5

A has * as in Figure 3 B, D, but is not mentioned in legend.

B, C show maternal genotype FL/D-tail, but is it easier to annotate as in previous figures the two indistinguishable genotypes than to suddenly label maternal genotype?

Figure 6A

The bottom Auxin - or + should better reflect which AID protein is degraded (corresponding to the top label "KNL-2 depletion or CENP-A depletion"). Otherwise, it is very confusing to just look at the bottom graph.

Line536. Typo: "maintans"

Line323. "S3A Fig" should be S4 Fig

Line 429-434. Consider rephrasing the summary of the findings "the CENP-A N-terminal tail is specifically required to re-establish correct centromere identity in......every generation" is more clear. Also "Once established, this centromere identity can then be inherited and maintained throughout development in the absence of the CENP-A N-terminal tail, but only......fails." to make it more clear.

---

## [Editor Report · Decision Letter 3]

28 May 2021

Dear Dr Steiner,

On behalf of my colleagues and the Academic Editor, Iain Cheeseman, I am pleased to say that we can in principle offer to publish your Research Article entitled "Trans-generational inheritance of centromere identity requires the CENP-A N-terminal tail in the C. elegans maternal germ line" in PLOS Biology, provided you address any remaining formatting and reporting issues. These will be detailed in an email that will follow this letter and that you will usually receive within 2-3 business days, during which time no action is required from you. Please note that we will not be able to formally accept your manuscript and schedule it for publication until you have made the required changes.

PRESS

Thank you again for supporting Open Access publishing. We look forward to publishing your paper in PLOS Biology. 

Sincerely, 

Ines

--

Ines Alvarez-Garcia, PhD 

Senior Editor 

PLOS Biology